# High-performance bifacial perovskite solar cells enabled by single-walled carbon nanotubes

Jing Zhang[1,10], Xian-Gang Hu [2,3,10], Kangyu Ji [4], Songru Zhao[5], Dongtao Liu[1], Bowei Li[1], Peng-Xiang Hou [2] ✉, Chang Liu [2], Lirong Liu[5], Samuel D. Stranks [4,6], Hui-Ming Cheng [1,2,7,8] ✉, S. Ravi P. Silva [1,9] ✉ & Wei Zhang [1,9] ✉

Bifacial perovskite solar cells have shown great promise for increasing power output by capturing light from both sides. However, the suboptimal optical transmittance of back metal electrodes together with the complex fabrication process associated with front transparent conducting oxides have hindered the development of efficient bifacial PSCs. Here, we present a novel approach for bifacial perovskite devices using single-walled carbon nanotubes as both front and back electrodes. single-walled carbon nanotubes offer high transparency, conductivity, and stability, enabling bifacial PSCs with a bifaciality factor of over 98% and a power generation density of over 36%. We also fabricate flexible, all-carbon-electrode-based devices with a high power-per-weight value of 73.75 W g$^{-1}$ and excellent mechanical durability. Furthermore, we show that our bifacial devices have a much lower material cost than conventional monofacial PSCs. Our work demonstrates the potential of SWCNT electrodes for efficient, stable, and low-cost bifacial perovskite photovoltaics.

Electricity is the most versatile and widely used form of energy in the world, and it is essential for decarbonization. Solar energy has the potential to supply abundant and affordable energy for various applications. To decarbonize the world, we need cheap and sustainable ways to generate electricity from solar power, which is abundant but underutilized[1]. Solar photovoltaic (PV) systems have grown rapidly in the past decade and are expected to continue to expand[2]. Over 60% of all current new energy world provision depends on solar energy. To achieve sustainable PV manufacture and deployment at gigawatt and terawatt scales, we need to lower the levelized cost of electricity (LCOE) of all existing and emerging PV technologies. The current pace of cost reduction is not sufficient for this aim. For instance, previous reports state the LCOE of commercial silicon solar cells is about 7 US cents per kWh and that of perovskite solar cells is about 5 US cents per kWh, whereas the average electricity price in the UK at present is around 34p per kWh (April 2023)[3,4]. Therefore, it is critical to further

[1]Advanced Technology Institute (ATI), University of Surrey, Guildford, Surrey GU2 7XH, UK. [2]Shenyang National Laboratory for Materials Science, Institute of Metal Research, Chinese Academy of Sciences, Shenyang 110016, PR China. [3]Advanced Interdisciplinary Research Center for Flexible Electronics, Academy of Advanced Interdisciplinary Research, Xidian University, Xi'an 710071, PR China. [4]Cavendish Laboratory, University of Cambridge, 19 J J Thomson Avenue, Cambridge CB3 0HE, UK. [5]Centre for Environment and Sustainability, Thomas Telford (AA) building, University of Surrey, Guildford, Surrey GU2 7XH, UK. [6]Department of Chemical Engineering & Biotechnology, University of Cambridge, Philippa Fawcett Drive, Cambridge CB3 0AS, UK. [7]Faculty of Materials Science and Energy Engineering, Shenzhen University of Advanced Technology, 291 Louming Road, Shenzhen 518107, PR China. [8]Shenzhen Key Lab of Energy Materials for Carbon Neutrality, Shenzhen Institute of Advanced Technology, Chinese Academy of Sciences, 1068 Xueyuan Road, Shenzhen 518055, PR China. [9]State Centre for International Cooperation on Designer Low-carbon & Environmental Materials (CDLCEM), School of Materials Science and Engineering, Zhengzhou University, Zhengzhou 450001, PR China. [10]These authors contributed equally: Jing Zhang, Xian-Gang Hu. ✉e-mail: pxhou@imr.ac.cn; cheng@imr.ac.cn; s.silva@surrey.ac.uk; wz0003@surrey.ac.uk

increase the power output per unit area of solar devices at low extra costs to accelerate the deployment of PVs. Bifacial PVs have the potential to harvest direct and diffuse sunlight from both front and back sides, leading to increased power output compared to monofacial PVs[3]. One advantage of bifacial PV technology is its independence from the angle of incident light (Fig. S1a, b, c)[5]. Previous studies have demonstrated that albedo energy, which is reflected from surrounding surfaces such as concrete or grassland, accounts for more than 20% of the standard AM 1.5 G irradiance (Fig. S1d–k). Related simulations and calculations have presented that the energy output of a bifacial PV module can be up to 30% higher than that of its monofacial counterparts[6]. Thus, bifacial PV modules have the advantage of being adaptable to any installation angle, and hence reduce the installation cost compared to monofacial PV modules which typically require costly solar trackers[7]. In this regard, bifacial PVs could offer a promising approach for further reducing the LCOE of PVs.

Bifacial PV devices can harvest albedo energy from the environment, which is measured by a "bifaciality factor (%, BiFi)", the ratio of their back- and front-side power conversion efficiency (PCE) under standard illumination conditions ranging from 0 to 1[3]. For instance, a BiFi of 80% implies that the back side can generate 80% of the power that the front side can produce. A higher BiFi means a higher utilization of the back side irradiance and a higher overall power output[8]. In the current bifacial PV market, crystalline silicon solar cells (c-Si) are dominant[9-11]. c-Si PVs have achieved modest-to-high BiFi (0.75–0.95) and high PCEs (over 24% for bifacial Si-cells), leading to their dominance in the market[11,12]. A notable example is the c-Si heterojunction with an intrinsic thin layer (HIT) solar cell, which has a BiFi of 93% and a PCE of 24.7%[13]. Other types of PVs, such as cadmium telluride (CdTe), copper indium gallium selenide, organic PVs and dye-sensitized solar cells, have received limited attention for bifacial applications[12]. The major reason is their comparatively low BiFi (e.g., the BiFi for CdTe is 15%) compared to c-Si PVs[14].

Fortunately, PSCs possess several physical parameters that make them suitable for bifacial PV applications. Previous studies have demonstrated that PSCs can achieve a high BiFi of ~94% (monofacial PCE is ~14%)[12]. We note that a record PCE of over 26% for single-junction mono-facial PSCs has been achieved[15,16], positioning them as a potential candidate for efficient bifacial devices. Furthermore, PSCs have an advantage over silicon solar cells under low-intensity light conditions[17], as they exhibit higher open-circuit voltage and lower voltage loss. This allows them to utilize the albedo energy more effectively.

For optimal bifacial applications, both front and back electrodes should be optically transparent, chemically stable, and compatible with the adjacent layers. Transparent conducting oxides (TCOs), such as indium tin oxide (ITO) and fluorine-doped tin oxide (FTO), are widely used for both front and back electrodes in bifacial PSCs. TCOs are not only highly conductive and transparent, but also resistant to halide corrosion and ion diffusion, enhancing the stability of bifacial PSCs[18]. For instance, a recent work reported that using FTO and indium zinc oxide (IZO) as front and back electrodes in bifacial PSCs resulted in an excellent performance of ~22%[19]. Besides, ITOs were employed as both front and back electrodes in minimodules and realized a great efficiency of 20.2% with an aperture area of 21.7 cm[2,20]. However, ITO and FTO as the front electrode TCOs suffer from brittleness which significantly restricts their application in flexible and foldable devices. Moreover, the fabrication of TCOs as back electrodes is challenging because the high temperature and plasma effects during deposition can damage the underlying perovskite layers[21]. Several solutions have been proposed to mitigate the damage caused by ITO or FTO deposition, including inserting a buffer layer on top of a perovskite layer or using damage-free sputtering approaches[22-24]. These methods increase complexity, the number of required layers of devices.

Other materials such as conductive polymers, metal grids, metal nanowires and carbon-based materials have also been proposed as alternatives to back electrodes in PSCs[25-28]. Among these materials, the integration of carbon nanotubes (CNTs) into perovskite-based devices stands out owing to the exceptional electrical conductivity, stability, and mechanical flexibility of CNTs[26,29-33]. These properties make them promising candidates for thermo-mechanical stable and flexible PSCs. Previously, graphene was used as the back electrode of PSCs and the resulted in devices with an efficiency of 18.6% (monofacial efficiency) with enhanced stability against heat and moisture[34]. CNTs have been also used as the front electrodes of flexible PSCs and enhanced their bendability and environmental stability[26]. These examples showcase the remarkable compatibility between carbon materials and PSCs, enabling them to function effectively as either front or back electrodes. In the past, carbon materials faced some challenges in bifacial PSCs. One limitation has been the comparatively low conductivity of thin and transparent carbon materials, which contributes to electrical losses within the devices. This issue is particularly prominent when considering their implementation as front electrodes. Another challenge includes their fabrication methods (e.g., water-soluble process)[35], which may complicate the requirements of the back electrode application or flexible device[35,36].

Here, we demonstrate the fabrication of bifacial PSCs using highly transparent and conductive single-walled carbon nanotubes (SWCNTs) as both front and back electrodes without integrating with other materials. We investigate the interactions between SWCNTs and electron/hole transport materials (ETM/HTM) and their effects on the device performance. These resulting bifacial PSCs achieve an ultra-high power generation density (PGD) of over 36 mW cm$^{-2}$. We adopt the BiFi notation to denote the corresponding PGD in a scientific and simple manner. For instance, a PGD of 25 mW cm$^{-2}$ BiFi$_{200}$ implies that the device produces 25 mW per cm$^2$ under simultaneous irradiance of 1-sun (1000 W m$^{-2}$) at the front and 0.2-sun (200 W m$^{-2}$, denoted as BiFi$_{200}$) at the back[37]. We further evaluated the photovoltaic performance of these bifacial PSCs under various common environments with different albedo factors and the resultant PGD range from 16.75 mW cm$^{-2}$ (tiles, BiFi$_{120}$) to 36.21 mW cm$^{-2}$ (snow, BiFi$_{960}$). Due to the excellent environmental durability of SWCNTs, these bifacial PSCs exhibit remarkable stability against moisture, light and heat conditions. We also simulated the power generation of these bifacial PSCs for the next 25 years. Moreover, SWCNTs are flexible and mechanically robust, which enables them to withstand repeated bending cycles. These features make SWCNTs suitable for flexible PSCs and enable new visions for the development of bifacial PSCs.

## Results and discussion
### Optoelectrical characterizations of SWCNT films
The SWCNT films were prepared using an injection floating catalyst chemical vapor deposition (FCCVD) method[38]. By controlling the collection time, SWCNT films of different thicknesses were obtained[38]. The preparation process is described in the *Methods*. The entangled SWCNT film with small bundles was observed using scanning electron microscopy (SEM) and transmission electron microscopy (TEM) (Fig. 1a, b). The mean diameter of each tube was determined to be ~2.2 nm (Fig. S2). The optical characteristics of the SWCNT films were investigated, revealing an impressive optical transmittance of ~95% at a wavelength of 550 nm (Fig. 1c). Notably, the optical transmittance in the visible wavelength range exceeds that of commercial ITO substrates, enabling the sufficient transmission of sunlight to perovskite absorbers. Unlike ITO, which shows a decrease in transmittance in the long wavelength region (>1000 nm), the SWCNT films preserve their transparency across these regions (Fig. 1c). This exceptional optical property renders the SWCNT films highly suitable as front electrodes for optoelectronic applications that encompass a broad spectrum of wavelengths. However, the sheet resistance of the SWCNT film with 95% optical transmittance (at 550 nm wavelength, denote as SWCNT@95%, ~138 Ω sq$^{-1}$) was higher than that of the commercial

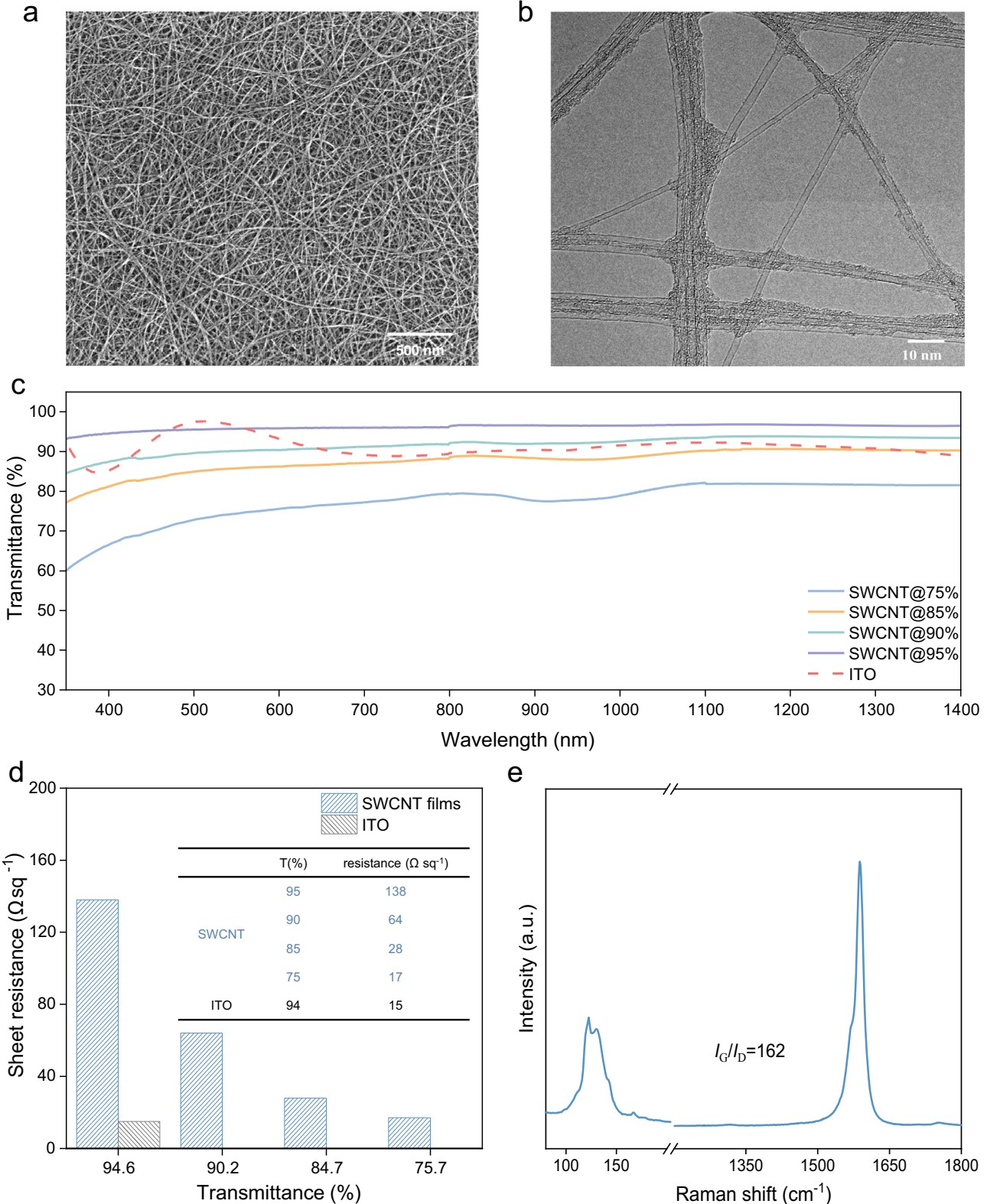

**Fig. 1 | Optical and electrical characterizations of SWCNTs prepared by FCCVD method. a** SEM image of the SWCNT film (scale bar = 500 nm). **b** TEM graph of a SWCNT film, and the average diameter of the individual tube is ~2.5 nm (scale bar = 10 nm). **c** Optical transmittances of the SWCNT films with various thicknesses. **d** Sheet resistances of SWCNT films. **e** Raman spectrum of the SWCNT film, and the high $I_G/I_D$ ratio is illustrated.

indium tin oxide (ITO, ~15 Ω sq$^{-1}$, Fig. 1d). To balance the sheet resistance and optical transmittance, we increased the thickness of SWCNTs and prepared SWCNT@90%, SWCNT@85% and SWCNT@75% films (~8 nm for SWCNT@95%, ~15 nm for SWCNT@90%, ~21 nm for SWCNT@85% and ~28 nm for SWCNT@75%)[39], and we found that the SWCNT@75% film has a sheet resistance comparable to that of the ITO film. We also compared the figure of merit values of these SWCNTs and found that the SWCNT@85% film presents the

highest value ($7.03 \times 10^{-3}$) among these SWCNTs (Table S1), which is even slightly higher than of ITO ($5.07 \times 10^{-3}$)[40,41]. The quality of the SWCNT films was then studied through Raman spectroscopy. A high-intensity ratio of G to D bands ($I_G/I_D = 162$) indicated high crystalline quality and low defect density, which facilitates the transportation of collected charges (Fig. 1e)[42,43]. To investigate the electrical properties of these SWCNT films further, we performed optical absorbance spectroscopy and observed the first and second semiconducting transitions of the van Hove singularities ($E^S_{11}$ and $E^S_{22}$), as well as the first metallic transition of the van Hove singularity ($E^M_{11}$) (Fig. S3). Typically, the SWCNT films prepared by a FCCVD method exhibit 1/3 metallic and 2/3 semiconducting characteristics[44]. The work function of the electrode is a key factor in enabling a high built-in voltage[45]. Therefore, the work function of SWCNT film was determined using ultraviolet photoelectron spectroscopy. The results indicate that the calculated work function of the SWCNT film is about −4.70 eV (Fig. S4). We note that the work function of the SWCNT film, which is comparable to that of ITO, remains constant at approximately −4.70 eV regardless of its optical transmittance[46–48]. Overall, this finding suggests that SWCNTs could be a viable alternative to ITO in terms of work function, sheet resistance and optical transmittance.

## Interactions between SWCNTs and charge transport materials

Here, we used phenyl-C61-butyric acid methyl ester (PCBM) and SnO$_2$ as the ETM. This combination can not only reduce the charge recombination at the perovskite/ETM interface, but also offer better electron mobility than a single SnO$_2$ or single PCBM[49]. The exterior layer that directly contacts the SWCNT film is PCBM, so we investigated the interaction between PCBM and SWCNTs here. The intimate contact between SWCNTs and PCBM was directly observed through TEM. The TEM analysis, as depicted in Fig. S5, along with the Fourier transform infrared spectroscopy data in Fig. S6, confirms the close interaction between SWCNTs and PCBM[39,50,51]. Raman spectra further revealed that the addition of PCBM to a SWCNT film slightly shifts the G band to lower wavenumbers, indicating an n-type doping behavior of PCBM on SWCNTs (Fig. S7)[52,53]. This conclusion contradicts some previous studies in which it was claimed that PCBM extracts electrons from the SWCNT films[54,55]. However, the previous studies focused solely on the interaction between PCBM and semiconducting SWCNTs, and did not consider the effect of PCBM on mixed SWCNTs, which are used in this work. In contrast to pure semiconducting or metallic SWCNTs, a mixed SWCNT film offers the advantage of harnessing the exceptional charge extraction capability of semiconducting SWCNTs, combined with the high conductivity provided by metallic SWCNTs. This unique combination ensures the efficient separation of electron-hole pairs and facilitates effective charge transport at the interface[56]. To further investigate the impact of PCBM on mixed SWCNTs, UV–vis spectra of SWCNT and PCBM-SWCNT were then analyzed. The results reveal a blue shift of the characteristic peak of SWCNTs, indicating strong π–π stacking interactions between PCBM and the SWCNT films (Fig. S8)[57]. And the work function of the SWCNT film decreased from ~4.70 eV to ~4.55 eV (Fig. S9). Moreover, the resistance of the SWCNT film reduced from 28.8 Ω to 21.9 Ω upon contact with PCBM, presumably owing to the formation of π–π bonds between PCBM and SWCNTs, which facilitated electron transport (Fig. S10)[50,58].

We subsequently studied the hole collection ability of SWCNT films. Here, copper-doped nickel oxide (Cu:NiO$_x$) was used as a hole transport material (HTM) due to its suitable work function (−5.3 eV), high hole mobility ($1.09 \times 10^{-2}$ cm$^2$ V$^{-1}$ s$^{-1}$), and simple fabrication protocols[26,59]. As revealed by previous studies, Cu:NiO$_x$ nanoparticles can optimize the morphology of the SWCNT films, while organic HTMs (e.g., PTAA) are too thin to compensate the rough surface of SWCNT films[26].

We then investigated the capability of extracting and transporting electrons within perovskite solar cells (PSCs) with SWCNT films (the ability to extract and transport holes for SWCNTs has been

investigated in our previous work)[26], semi-devices with a configuration of glass/perovskite/SnO$_2$/PCBM/SWCNT (with various optical transmittances, 95%, 90%, 85% and 75%) were prepared, and steady-state photoluminescence (PL) and corresponding time-resolved photoluminescence (TRPL) were then conducted. Compared to the bare perovskite thin film, the PL intensity showed a significant reduction for all SWCNT-based semi-devices (Fig. 2a, the inset picture shows the configuration of the semi-device), demonstrating similar electron extracting capability of ETM/SWCNT stacks with slight variation for semi-devices based on SWCNT films with different optical transmittances. To quantify this variation among devices, TRPL data were illustrated and calculated (Fig. 2b). As a result, the decay time decreased from 11.39 ns to 4.61 ns as the optical transmittance of SWCNT films reduced from 95% to 75%, whereas the values were 9.63 ns and 6.72 ns for semi-devices with SWCNT@90% and SWCNT@85% respectively[60]. In conjunction with the PGD of resultant devices (Table S3), this indicated that thick SWCNT films could collect charges more efficiently than their thinner counterparts[37,61]. In addition to the difference in conductivity of these SWCNT films, it is plausible that the compactness of the contact between PCBM and thick SWCNT films may contribute to the observed reduction in carrier lifetime[62], which is consistent with the PL data. To further investigate the influence of the SWCNT films on charge transport behavior, carrier lifetime mapping was performed by comparing the lifetime of charges in areas with and without a SWCNT film. The transfer process of the SWCNT films can be found in Fig. S11. The half-stack device configuration (glass/perovskite/SnO$_2$/PCBM/with and without SWCNT@85%) is illustrated in Fig. S12, and the PL mapping results reveal that the average lifetime of carriers is reduced from 1.44 ns (blue box, 6 μm × 6 μm) for glass/perovskite/SnO$_2$/PCBM to 0.94 ns (orange box, 6 μm × 6 μm) for glass/perovskite/SnO$_2$/PCBM/SWCNT@85% (Fig. 2c). This difference in lifetime provides evidence that the carriers in the perovskite/ETM/SWCNT stack travel more quickly than those in the perovskite/ETM stack, suggesting that additional electron pathways may be formed at the PCBM/SWCNT interface, which leads to the observed reduction in lifetime.

## Characterizations of all-carbon-electrode-based bifacial PSCs

We then fabricated bifacial p-i-n devices using an glass/SWCNT@85%/Cu:NiO$_x$/Perovskite/SnO$_2$/PCBM/SWCNT architecture and measured their corresponding performances using eight common ground reflection materials under AM 1.5 G incidence. The choice of SWCNT@85% as the front electrode was due to its balanced conductivity and optical transmittance[26]. The reflection spectra and albedo factor of the eight materials can be found in Fig. S13 and Fig. S14. We then summarized the performances of these all-carbon-based bifacial PSCs under various situations in Fig. 3a and Table S2. It can be seen that PSCs with a back electrode of SWCNT@75% exhibit the highest PGD when illuminated from the front side under 1 sun (19.14%, PGDs of other devices can be found in Tables S3). When considering albedo energy, bifacial devices equipped with SWCNT@85% as both front and back electrodes (denoted as double-sided SWCNT@85% devices) deliver the highest efficiencies among all four structures under high albedo conditions (albedo > 20%), whereas monofacial PSCs do not efficiently utilize this albedo energy (the photovoltaic parameters of a monofacial PSC illuminated from the metal-electrode side are shown in Fig. S15). The remarkable PGD in the double-sided SWCNT@85% PSC can be attributed to its exceptional BiFi. Among recently reported efficient bifacial PSCs, the double-sided SWCNT@85% device exhibits the highest BiFi, reaching an impressive value of 98.27% (Table S4), while the BiFi for devices incorporating back electrodes of SWCNT@95%, SWCNT@90%, and SWCNT@75% were measured to be 93.36%, 94.25%, and 74.24%, respectively.

To further confirm the reliability of short-current density ($J_{sc}$) data, we collected external quantum efficiency (EQE) spectra for the

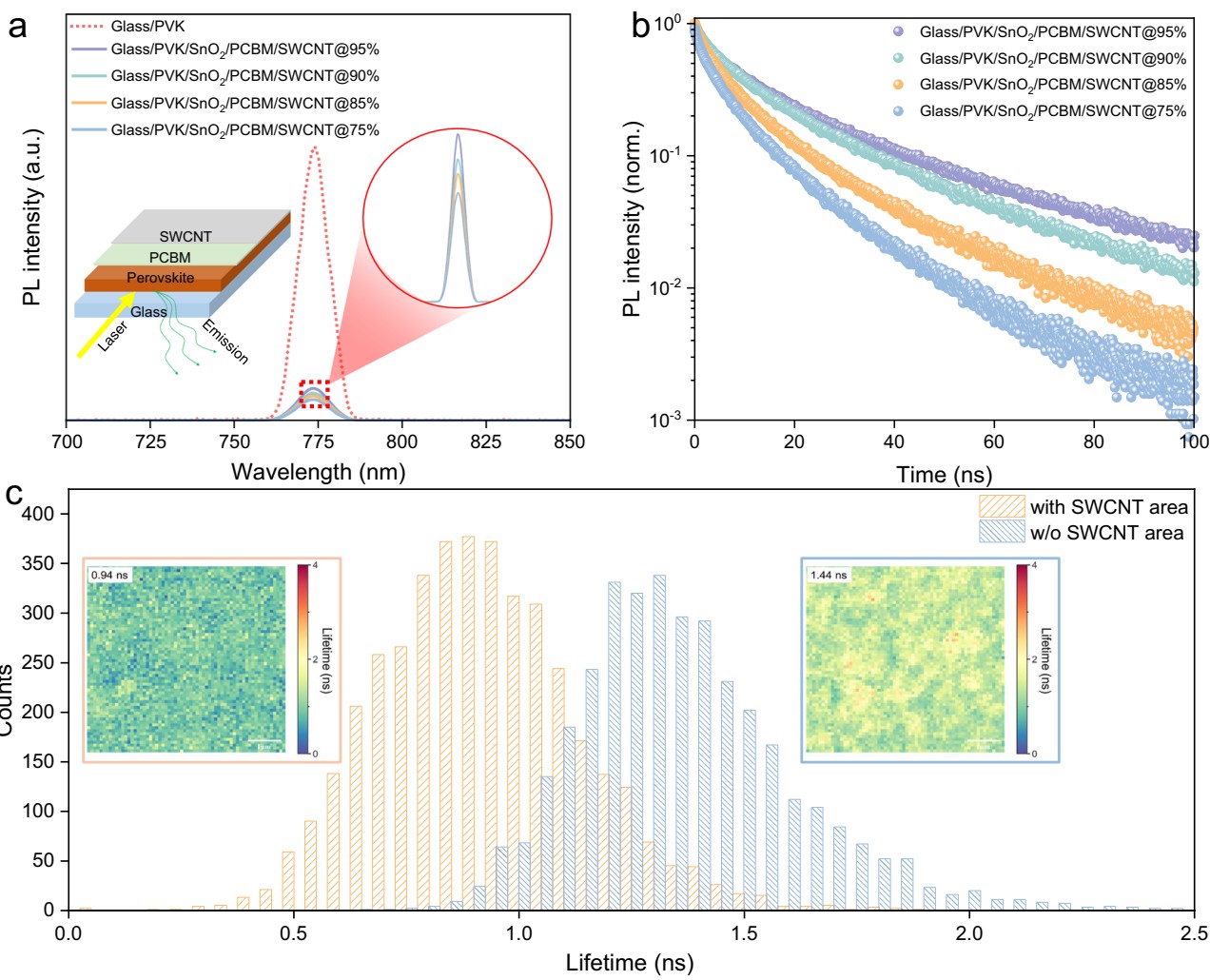

**Fig. 2 | Optical characterizations of semi-devices. a** Steady-state PL of semi-devices with various SWCNT films. **b** Corresponding TRPL curves of semi-devices with SWCNT films in different thicknesses. **c** Charge lifetime statics of areas with and without SWCNTs (the fluence power is 0.4 μJ cm$^{-2}$), and the inset pictures are charge lifetime mapping of areas with and without SWCNT films (the size of the area is 6 μm × 6 μm), with average lifetime marked on the top left.

devices with different SWCNT films (refer to Fig. S16), and these values are in agreement with the trend for $J_{sc}$ shown in Table S4. Moreover, the performance of the bifacial PSCs was assessed using double light source protocols (refer to Fig. 3b), as outlined in the International Electrotechnical Commission Technical Specification 60904-1-2[11,31]. The setup of the test is presented in the inset of Fig. 3b, where the device receives continuous AM 1.5 G one sun illumination from both the front and back sides simultaneously, enabled by the mirror design. As a result, the double-sided SWCNT@85% device exhibits an impressive $J_{sc}$ of ~40 mA cm$^{-2}$, resulting in an equivalent PGD reaching 36.51 mW cm$^{-2}$ BiFi$_{1000}$, suggesting the potential of bifacial PSCs to enhance equivalent bifacial-PGD (18.54% and 18.22% for frontside and backside) performance by utilizing surrounding reflected irradiance. In contrast, under the same condition, the opaque control cell has a bifacial PGD of 23.63 mW cm$^{-2}$ BiFi$_{1000}$, which is only ~0.3% higher than that of one-side illumination (23.38%, Fig. S17). To directly ascertain the degradation processes occurring in perovskite films, we conducted long-term stability tests on these devices without encapsulation under ~55% relative humidity (RH). For comparison, we used a control cell with a configuration of ITO/Cu:NiO$_x$/Perovskite/SnO$_2$/PCBM/BCP/Ag. The PGD of the control cell dropped from 23.38% to 13.52% (the initial $J$-V data of the control cell can be found in Fig. S18, unencapsulated), corresponding to 30% of its initial PGD after 30 days. By removing both the metal and ITO electrode, utilizing the excellent hydrophobic

properties of the SWCNT films, the double-sided SWCNT@85% device retained over 95% of its original PGD after 30 days (Fig. 3c, temperature = 25 °C, RH = 55%, dark, unencapsulated). In practical situations, thermal stability is as critical for bifacial devices because they harvest energy from both sides, leading to higher operating temperatures than their monofacial counterparts[63]. The PGD of a double-sided SWCNT@85% PSC dropped from 18.55% to 16.63%, maintaining around 90% of their initial efficiencies even after being exposed to 85 °C over 30 days (Fig. 3d). Conversely, the control device did not survive the test and expired after around 3 weeks. Additionally, the encapsulated device retained about 90% of its initial PGD after 30 days under continuous light illumination at the maximum power point (MPP, 100 mW cm$^{-2}$, AM 1.5 G, 25 °C), without an ultraviolet filter. In contrast, the encapsulated control device only retained ~60% of its original values (Fig. 3e). Thus, these results demonstrate the potential of SWCNT electrodes for efficient and stable PSCs.

The mild and simple transfer process of SWCNT films confers remarkable compatibility with flexible substrates, enabling the realization of bifacial flexible devices. We transferred the device configuration to flexible substrates and achieved excellent PCEs of 17.1% (front side) and 16.6% (back side) with a PEN/SWCNT@85%/Cu:NiO$_x$/Perovskite/SnO$_2$/PCBM/SWCNT@85% configuration (Fig. 4a, photovoltaic parameters can be found in Table S5), obtaining a high BiFi of 96.90%, which is the highest value among flexible bifacial solar cells

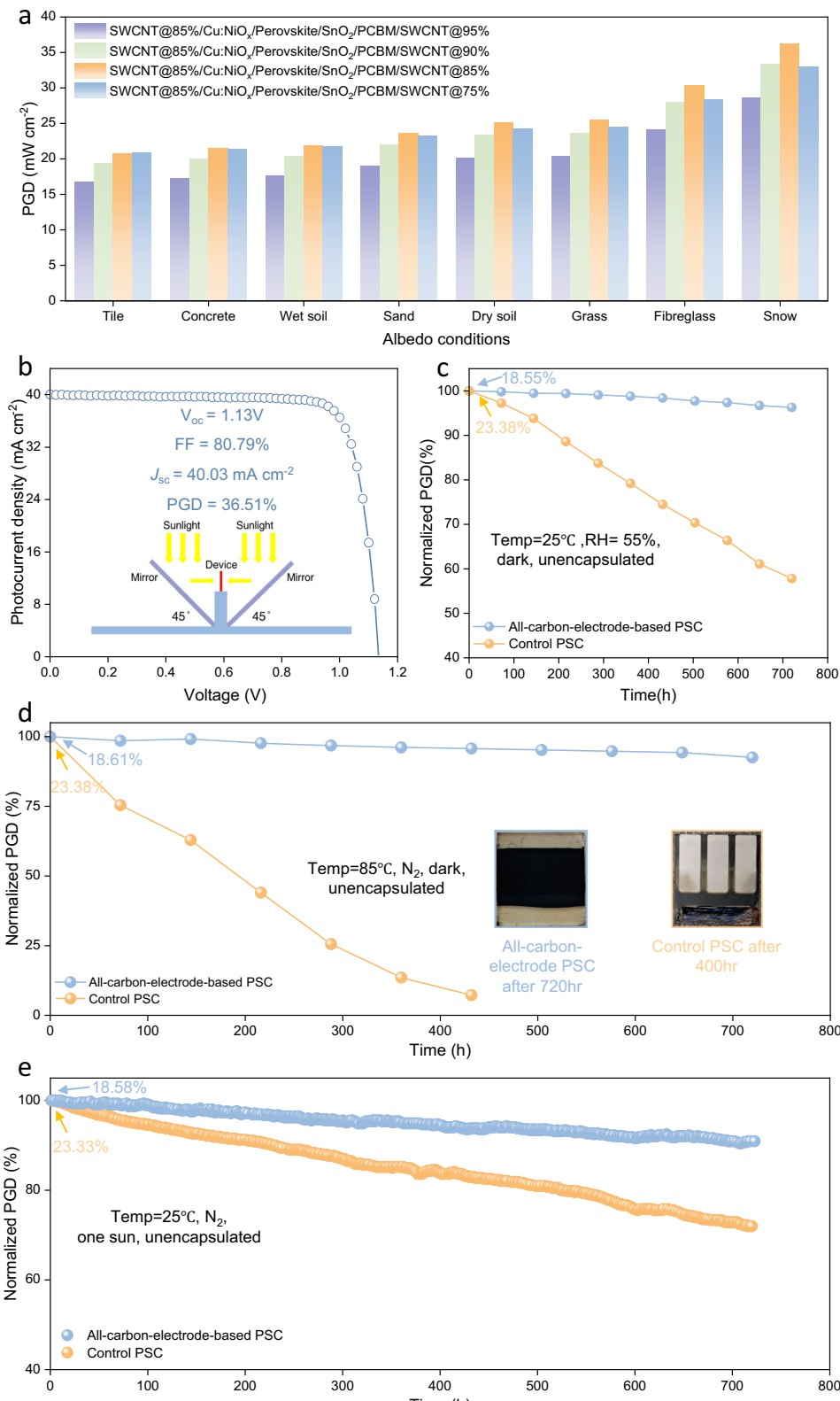

**Fig. 3 | Photovoltaic performances of the all-carbon-electrode-based bifacial PSCs. a** PGD chart of bifacial PSCs under various albedo conditions. **b** *J*-V curves of an all-carbon-electrode-based PSC illuminated from the front side (100 mW cm⁻²) and back side (100 mW cm⁻²) simultaneously, and the inset picture depicts the characterization setup described in IEC TS 60904-1−2: double light source method. **c** Shelf storage stability records of an all-carbon-electrode-based PSC and control device under 25 °C and a relative humidity of 55% (dark, unencapsulated). **d** The thermal stability performance of all-carbon-electrode-based PSCs and their control counterparts were evaluated under sustained exposure to 85 °C in an N₂-filled glove box, and inset pictures show that the control PSC becomes yellowish after 400 h (with orange border) whereas the all-carbon-electrode-based PSC is still black (with blue border). **e** Maximum power point (MPP) tracking of an all-carbon-electrode-based PSC and control device under N₂ atmosphere, 1 sun continuous illumination without UV filter. (Temperature = 25 °C).

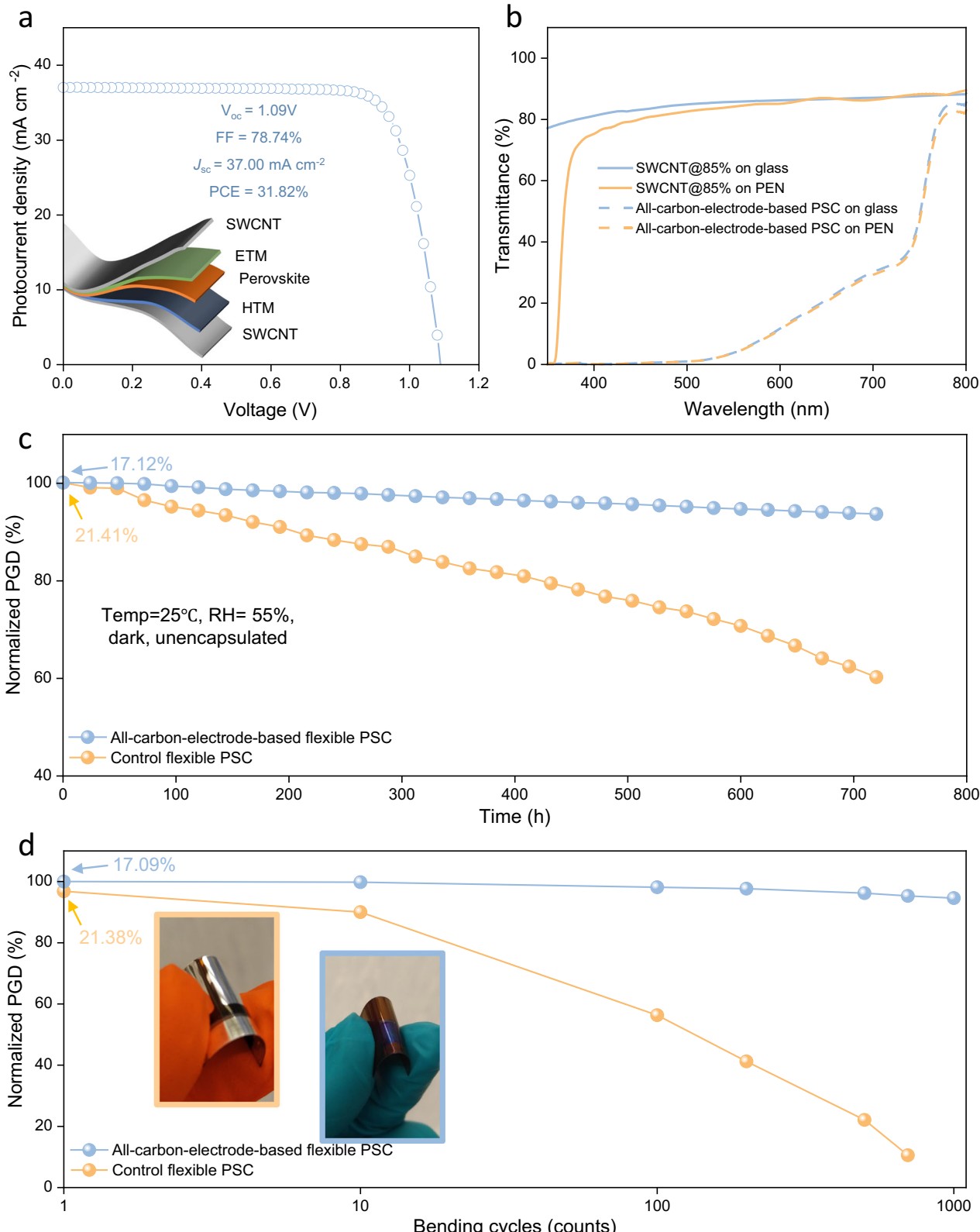

**Fig. 4 | Photovoltaic parameters and stability of flexible all-carbon-electrode-based PSCs. a** *J*-V curves and corresponding PGD of a flexible double-sided SWCNT@85% device measured by double light source method. **b** Optical transmittances of SWCNT films and corresponding devices on glass and PEN, respectively. **c** Shelf storage (in dark) stabilities of all-carbon-electrode-based flexible PSCs and controls devices (both unencapsulated) at a temperature of 25 °C and a relative humidity of 55%. **d** Bending performances of flexible devices (bending radius of 4 mm, 1000 cycles) and the inset pictures show the graphs of a flexible control device (with orange border) and an all-carbon-electrode-based flexible device (with blue border).

(Table S6). We also characterized the performance of the flexible double-sided SWCNT@85% PSCs under a double light source setup, and a high PGD of 31.82 mW cm$^{-2}$ BiFi$_{1000}$ was achieved. Because the flexible PSC is lightweight (~4.4 g m$^{-2}$), therefore, an exceptional power-per-weight of 73.75 W g$^{-1}$ was realized (without packaging, while the value for silicon is ~5 W g$^{-1}$), which is the highest value so far[64,65]. In comparison to their rigid counterparts, the slightly lower PGD can be attributed primarily to the loss of $J_{sc}$ (~21.0 mA cm$^{-2}$ for a rigid bifacial PSC and ~19.4 mA cm$^{-2}$ for a flexible bifacial PSC) caused by the lower optical transmittance of the flexible substrates (Fig. 4b). From the optical transmittance data, this loss is primarily in the wavelength range from 350 nm to 600 nm, which overlaps with the strong absorption range of the perovskite, resulting in optical losses in flexible PSCs. In terms of environmental stability, the water vapor transmission rate (WVTR) of plastic substrates is much higher than that of glass substrates (e.g., at 40 °C, the WVTR is ~7 g m$^{-2}$ day$^{-1}$ for PEN, while the value is ~10$^{-5}$ g m$^{-2}$ day$^{-1}$ for glass)[66], which is the main reason for the relatively short lifespan of flexible PSCs. In this study, we employed a method that sandwiches the perovskite layer between two hydrophobic SWCNT films, effectively reducing the amount of moisture that can reach the photoactive layer, thereby ensuring the prolonged lifetime of the flexible PSCs. Our all-carbon-electrode-based flexible PSCs demonstrated significantly improved storage lifetime, as shown in Fig. 4c, and the PGD of flexible double-sided SWCNT@85% device reduced from 17.1% to 15.9% after 1 month, reserving ~93% of their initial PGD, while the control device could only retain ~58% of its pristine value (whose front electrode is ITO and back electrode is Ag, and the initial photovoltaic parameters of flexible control devices can be found in Fig. S19). We performed a mechanical stability test on the flexible PSCs with all-carbon electrodes and the control devices by bending them with a 4 mm radius at a rate of 20 cycles/min in a glove box. We measured the PCE every 100 cycles for a total of 1000 cycles. The control device showed severe degradation, with the ITO cracking or breaking after 1000 bending cycles (Fig. S20), leading to a drastic rise in sheet resistance (Fig. S21). Conversely, the flexible double-sided SWCNT@85% PSC maintained over ~97% of its initial PGD (Fig. 4d), no obvious change on the morphology of SWCNT was observed (Fig. S22 a,b), and the sheet resistance remains constant (Fig. S21). Therefore, SWCNT films unlock a promising approach to achieve lightweight, highly-efficient, mechanically robust and long-term stable flexible bifacial PSCs.

## Cost analysis

Material cost is a critical factor for commercialization. To accurately assess the economic feasibility of different devices, cost analysis should rely on realistic material price and quantity used in fabrication, with minimal assumptions to ensure the validity of small-scale estimates. Here we consider that all devices are interconnected in series to form a module, following a common practice in thin film production[67]. We also assume that electricity and water consumption are identical for all devices and do not affect the cost comparison. As shown in Table S7 and S8, the material cost of an all-carbon-electrode-based bifacial PSC is about 70% lower than that of a monofacial device (whose electrodes are ITO and Ag). The cost advantage is further enhanced when accounting for equipment costs, as all-carbon-electrode-based devices eliminate the need for expensive radio frequency (RF) sputtering or thermal evaporation processes for ITO or metal electrodes. Therefore, our results indicate that all-carbon-electrode-based bifacial PSCs offer a promising route to achieve efficient, low-cost and long-term stable devices compared to monofacial devices that use ITO and Ag.

## Power generation simulation

We performed a simulation of the power generation of all-carbon-based bifacial PSCs under various albedo conditions, based on the photovoltaic results obtained from the previous section. We employed the Norwegian Earth System Model as the simulation tool, which predicted the weather and daytime period in England for the next 26 years (from 2025 to 2050). More details of the simulation method are provided in the *Methods* section. We used the double-sided SWCNT@85% PSCs as an example (simulation data of monofacial Si solar cells, bifacial devices with SWCNT@95%, SWCNT@90% and SWCNT@75% are shown in Fig. S23a–h). For the 1-year (2025) power generation, the power output could reach as high as ~35 kWh in the sunlight-rich month (Fig. 5a, grass, June). In contrast, the monofacial silicon cell could only generate ~28 kWh under similar conditions (Fig. S23a, grass, June). Moreover, for the winter month with a short daytime (e.g., January and December), the double-sided SWCNT@85% PSC could harvest albedo energy efficiently, producing ~13 kWh (Fig. 5a, grass, January) while ~11 kWh for silicon counterparts (Fig. S23a, grass, January). Regarding the long-term power generation (from 2025 to 2050), double-sided SWCNT@85% PSCs are projected to generate ~300 kWh per year of electricity within these 26 years (Fig. 5b, grass), whereas the monofacial silicon cell could only deliver around 260 kWh per year (Fig. S23b). These simulation results confirm the remarkable potential of bifacial PSCs with carbon materials, especially in terms of enhanced energy generation under various albedo conditions. This is particularly important in urban areas where the widespread use of glass increases the reflection and availability of incident sunlight.

# Methods

## Materials

ITO-patterned glass substrates (10 Ω sq$^{-1}$) were purchased from Huananxiangcheng Ltd. (China) and ITO-patterned PEN flexible substrates (<15 Ω sq$^{-1}$) were purchased from Peccell Inc (Japan). Nickel (II) nitrate hexahydrate (98%) was purchased from Alfa Aesar. Sodium hydroxide (99.9%) and pH paper stick indicator were purchased from Fisher Sci Ltd. Lead diiodide (PbI$_2$, 99.99%) and lead dibromide (PbBr$_2$, 99%) were purchased from Tokyo Chemical Industry Co., Ltd. (TCI, Japan). Formamidinium iodide (FAI), methylammonium iodide (MAI), Cesium iodide (CsI, 99.999%), copper (II) nitrate trihydrate (98%), nitric acid (HNO$_3$, 70%), and choline chloride (≥99%) were purchased from Sigma-Aldrich. PC$_{61}$BM (99.5%) and Bathocuproin (BCP, 98%) were purchased from Ossila. N,N-dimethylformamide (DMF, 99.8%), dimethyl sulfoxide (DMSO, 99.7%), chlorobenzene (CB, 99.8%) and Isopropanol (IPA, 99.5%) were purchased from Acros. SnO$_2$ (2.5 wt%) in butanol solution was purchased from Avantama. Filter papers were purchased from Whatman Inc. All chemicals and materials were purchased from commercial suppliers and used as received.

## SWCNT film preparation

SWCNTs films were prepared according to a floating catalyst CVD process[38,68]. And the SWCNT films were formed on a membrane filter at room temperature. The thickness of the SWCNT films was controlled by adjusting the collection time. Glass substrates and PEN substrates were cleaned with deionized water, acetone, and isopropyl alcohol respectively for later use. Then the membrane filter with the SWCNT film deposited first tailored to a proper size with a scalpel and placed on a pre-cleaned substrate. Then a drop of alcohol was dropped on the transferred SWCNT film to ensure good contact between the substrate and the SWCNT film. After several seconds the membrane filter was removed by using a tweezer. Finally, a glass/SWCNT or PEN/SWCNT substrate was obtained to be used to deposit PSCs. For HNO$_3$ doping, the glass/SWCNT or PEN/SWCNT substrate was transferred to a petri dish, and one drop of HNO$_3$ was dropped on top of the substrate, followed by drying under 60 °C for 45 min. Then these substrates were washed with diluted water and alcohol for later use.

## Calculation of the cost of SWCNT films

The raw materials and the energy used during the production (here is FCCVD) are taken into account. As calculated, the total cost of 1 gram SWCNTs is estimated at ~$110, The weight of a single SWCNT film (~85%

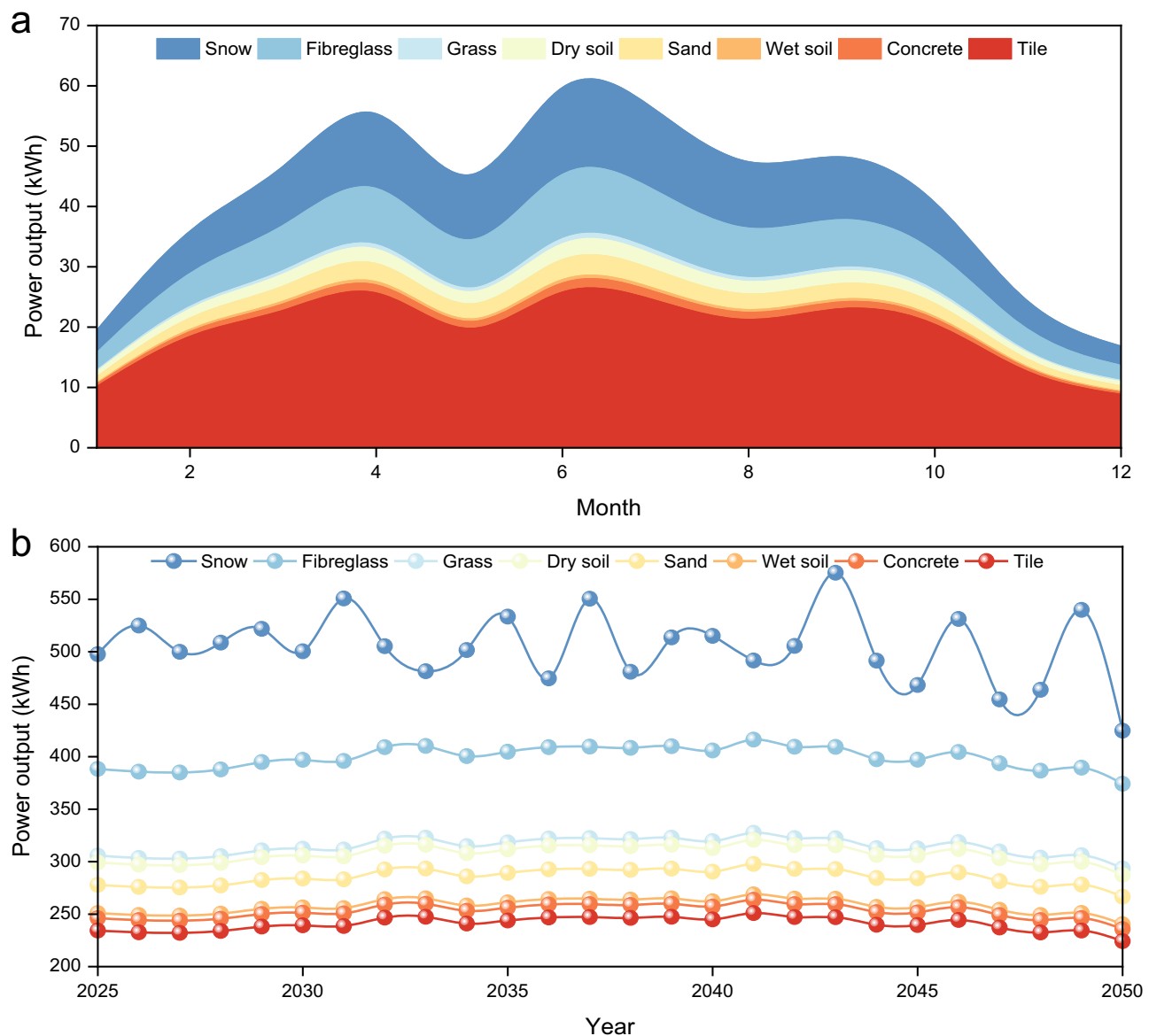

**Fig. 5 | Short-term (1 year) and long-term (26 years) power output simulation of double-sided SWCNT@85% PSCs. a** The 1-year power generation and the (**b**) power output for 26 years (from 2025 to 2050) (assuming the device area is 1 m²). (Source data are provided as a Source Data file).

optical transmittance) used in this work is ~460 µg. Therefore, the cost of each film is ~$0.0506. And the area of one single SWCNT film is 21.23 cm² (equal to 0.002123 m²). Thus, the cost of SWCNT is ~$24 m⁻². While comparing with ITO-coated glass substrates, the cost of glass substrates needed to be added. The final cost of glass/SWCNT is estimated at ~$40 m⁻².

**Copper-doped nickel oxide thin film preparation**
0.25 mol nickel (II) nitrate hexahydrate and copper (II) nitrate trihydrate were dissolved in 50 ml distilled water with continuous stirring, copper (II) nitrate trihydrate was added to the nickel (II) nitrate hexahydrate solution at a molar ratio of 5%. After the solution became dark green clear, the pH value was adjusted to 10 by adding 10 M sodium hydroxide solution, the pH value was measured by pH indicator. The solution was kept stirring for another 15 min after pH reached 10. Then the solution was filtered by a filter paper to collect the dark green precipitation. The collected precipitation was heated at 80 °C for 1 h after washing with distilled water three times. To obtain copper-doped nickel oxide nanoparticles (Cu:NiO$_x$ NPs), the heated precipitation was

transferred to a furnace to calcinate at 275 °C for 2 h to obtain black powders. Finally, the black powder was dispersed in deionized water to form 2 wt% Cu:NiO$_x$ NPs solution.

**Perovskite solar cell fabrication**
ITO-patterned glass substrates (10 Ω sq⁻¹, Huananxiangcheng Ltd.) and ITO-patterned PEN flexible substrates (<15 Ω sq⁻¹, Peccell) were cleaned with deionized water, acetone, and isopropyl alcohol respectively. Then ITO-patterned glass substrates were treated with an oxygen plasma process (Emitech K1050X, 230 V, 100 W) for 5 min before fabrication. The rest fabrication processes of ITO-based and SWCNT-based perovskite solar cells were the same. Cu:NiO$_x$ NPs solution was spin-coated on ITO-based and SWCNT-based substrates at 2000 rpm for 20 s, followed by a post-heating process at 120 °C for 10 min. Then, these Cu:NiO$_x$-coated substrates were moved to a UV-ozone cleaning device to receive 5 min of UV-Ozone treatment. Then the KI layer was prepared on top of the Cu:NiOx via spin-coating, the spin-coating condition of the KI layer is 2000 rpm, 30 s with a solution

concentration of 2 mg/ml (in water), and heated at 100 °C for 10 min. Here, the composition of the perovskite layer is $Cs_{0.05}FA_{0.80}MA_{0.15}Pb(I_xBr_{1-x})_3$, the perovskite precursor solution was prepared by dissolving 470.23 mg $PbI_2$, 66.06 mg $PbBr_2$, 15.59 mg CsI, 166.64 mg formamidinium iodide, and 19.15 mg methylammonium bromide in a 1 ml solution of 4:1 V/V DMF/DMSO. Following this, the solution was stirred overnight at room temperature. Then the perovskite layer was formed by spin-coating according to a two-step protocol, 1000 rpm for 10 s and 4000 rpm for 35 s, 80 µl CB was dropped 5 s before the end of the second step. Then the film was heated at 100 °C for 60 min on a hotplate. A passivation layer, choline chloride was then fabricated on top of the perovskite layer at 4000 rpm, 30 s with a solution concentration of 1 mg/mL (in IPA), and heated at 100 °C for 30 min. The $SnO_2$ layer was prepared with the $SnO_2$ butanol solution, which was diluted with 1-butanol until the concentration reached 1.25 wt%. Subsequently, $SnO_2$ thin film was spin-coated on top of the perovskite with the diluted solution at 6000 rpm (2000 rpm/s) for 30 s and heated at 100 °C for 40 min. PCBM was employed as the electron transport material, $PC_{61}BM$ (20 mg/ml in CB) was spin-coated at 2000 rpm for 20 s, and dried at 100 °C for 5 min. As for the back SWCNT electrode, the transfer process is the same as the front SWCNT transfer process. For control cells, BCP solution (0.5 mg/mL in IPA) was spin-coated on top of $PC_{61}BM$. Finally, a 100 nm thick Ag electrode was deposited in a thermal evaporator ($>3 \times 10^{-6}$ Torr, Moorefield thermal evaporator) to complete the fabrication process.

## Materials and device characterization (SWCNT films and PSCs)

The optical transmittance of SWCNT film was measured by a UV–Vis–NIR spectrophotometer (AGILENT CARY 5000). Raman spectra excited by 633 nm laser was obtained using a Jobin-Yvon Labram HR800 instrument. The sheet resistance of the SWCNT film was determined by Four Probes Tech (4-probe Tech.). The morphology of the SWCNT film was characterized by SEM (Nova Nano SEM 430) and TEM (JEM-2010HR and Tecnai G2 F20, operated at 200 kV). XRD patterns were performed using 45 kV, 40 mA Cu Kλ ($\lambda = 0.154187$ nm) radiation by PANalytical X'Pert Pro, equipped with an X-ray mirror (parallel beam) and proportional Xe detector (GM-tube). The work function of SWCNT films was determined via Esclab-250 instrument. Photoluminescence data were carried out by Agilent Cary Eclipse Fluorescence Spectrophotometer. AFM images were obtained via AIST-NT SmartSPM 1000 in a tapping mode. The TRPL measurements were measured using a confocal microscope setup (PicoQuant, MicroTime 200). The sample was excited by a 510-nm pulsed diode (PDL 828, PicoQuant, pulse width ~100 ps) with an air objective. The signals were focused onto a Hybrid PMT detector connected to a Picoquant acquisition card for time-correlated single-photon counting (time resolution of 200 ps). The J-V characteristics and MPP tracking were performed outside the glove box at the lab condition by using a Keysight B2901A source meter under simulated one-sun AM 1.5 G illumination ($100 \, m^{-1} W^{-1} cm^{-2}$) with a AAA steady solar simulator (Enlitech, SS-F5-3A). Before J-V measurements, the simulator was cautiously calibrated by using a standard monocrystalline silicon reference solar cell (Fraunhofer ISE CalLab (ISE001/013-2018)) with a KG-5 filter. The sweeping conditions are reverse scan (1.20 V to −0.02 V, scan rate 40 mV s$^{-1}$, and no delay time), forward scan (-0.02 V to 1.20 V, scan rate 40 mV s$^{-1}$), with no delay time), All devices were measured both in the reverse scan (1.20 V to −0.20 V, step 0.02 V, delay time 100 ms) and forward scan (−0.20 to 1.20 V, step 0.02 V, delay time 100 ms) without any pre-light soaking and pre-bias process. To ensure accuracy, a mask with an aperture area of 0.09 cm$^2$ was employed during the measuring process[69,70]. The data were

collected by the IV testing system with software (IVS-KA5000). The stabilized power output was measured at the maximum power output bias voltage. EQE was measured in air on a commercial system (Bentham PV300). And the stability test was conducted without encapsulation. The stability test was performed at 25 °C, and the RH was ~55%. The repeated bending cycle tests are performed by a custom-made stretching machine with an adjustable bending radius.

## Models and simulations

The simulation is carried out by PVLib, which is an open-source solar generation simulation software based on Python[71]. The weather forecast data we use for this simulation, such as solar irradiance, temperature and near-surface wind speed, are generated by the Norwegian Earth System Model with lower atmosphere–land resolution under Shared Socioeconomic Pathways 1–2.6[72]. The geographic location of this simulation is set on the west coast of the UK (53.86 N 3.07 W). We assume the devices have an effective area of 1m$^2$, and they are fixed to the angle with maximum annual generation facing towards the south.

The Norwegian Earth System Model generates four weather forecast data (Surface Downwelling Shortwave Radiation, Surface Diffuse Downwelling Shortwave Radiation, Near-Surface Wind Speed, and Near-Surface Air Temperature) with a monthly time scale. The forecast data can be converted into ideal hourly data for 1 day of that month by referring to historical hourly data. This hourly data is then fed into the PVlib simulation model to simulate the generation of that day. The effective irradiance of solar cells consists of three irradiances, beam irradiance, ground-reflected irradiance and sky-diffuse irradiance. The albedo is defined as ground surface reflectivity, and it directly determines the ground-reflected irradiance. With different albedo, the PGD of the bifacial perovskite cells varies as well as effective irradiance changes. The generation of the cell is simulated by the effective irradiance it receives. Since the devices are in the lab stage, the losses caused by the package are neglected and the temperature coefficient for all perovskite solar cells is assumed to be −0.1%/°C.

## Reporting summary

Further information on research design is available in the Nature Portfolio Reporting Summary linked to this article.

# Data availability

The main data supporting the findings of this study are available within the published article and its Supplementary Information and source data files. Additional data are available from the corresponding author on request. Source data are provided with this paper.

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

## Acknowledgements

J.Z., W.Z. and S.R.P.S thanks the EPSRC standard research (EP/V027131/1), Newton Advanced Fellowship (192097) and Equal Opportunities Foundation Hong Kong for financial support for financial support. X.G.H. thanks the National Natural Science Foundation of China (62304163) and Qinchuangyuan Cited High-level Innovation and Entrepreneurship Talent Projects (QCYRCXM–2022-364). HM.C. acknowledges support by the National Natural Science Foundation of China (No. 52188101), K.J. and S.D.S. acknowledge funding from the Royal Society and Tata Group (UF150033) and support from the European Research Council (European Union's Horizon 2020, HYPERION 756962 and PEROVSCI 957513).

## Author contributions

These authors contributed equally: J.Z., XG.H. XG.H. synthesized and collected SWCNT films, and transferred SWNCT films to substrates with the supervision of PX.H., C.L. and HM.C. J.Z. fabricated perovskite solar cells. B.L. and D.L. performed J-V curve, EQE, and MPPT characterizations. K.J. and S.D.S carried out the PL, TRPL, and charge lifetime mapping measurement. S.Z. and L.L. did the power output simulation. J.Z. drafted the paper. PX.H., HM.C., S.R.P.S, and W.Z. supervised the research and contributed to the analysis and writing of the paper. All authors helped fine-tune the paper.

## Competing interests

The authors declare no competing interests.
