## [Peer Review File · Nature Communications]

High-performance bifacial perovskite solar cells enabled by single-walled carbon nanotubesReviewer #1 (Remarks to the Author):

Recommendation: Minor revision.

Comments:

In this manuscript, Zhang et al., reported an interesting bifacial PSCs using highly transparent and conductive SWCNTs. Although the author delightedly reported a high PCE over 36%, this result was measured under the designated albedo factors and an active area of 0.09 cm², which somehow misleads the readers. It should be highlighted that the flexible device also showed excellent performance. Therefore, this work should be a milestone for the bifacial PSCs using SWCNTs, which definitely gives a general interest, and another way for future development of bifacial SWCNTs. This work could be accepted for publication in Nat. Commun. after revising the unclear points listed below:

- 1) It is true that SWCNTs have better optical transmittance. But it is inaccurate to mention the high cost using back metal electrodes compared with SWCNTs. The author should revise this description, e.g., in Abstract and Introduction part.
- 2) In the discussion on working function of SWCNTs by UPS in Fig.S2, did author mean all SWCNTs films with different transmittance showed same working function? In addition, although SWCNTs electrodes mainly consisted of metallic tubes, there should remain some semiconductive tubes, which influences the transmittance and working function. Therefore, the author is recommended to conduct the UV-vis-NIR to check the metallic/semiconductive property by observing the M11, S11 etc., (doi.org/10.1016/j.carbon.2023.01.011).
- 3) In Fig. 2b, the TEM showed that there is something (maybe surfactant) wrapped SWCNTs. Therefore, it is suspicious that Fig. S3 can conclude the PCBM surrounded the SWCNTs. If so, the author should collect more evidence to support Fig. S3, for instance compare by FTIR or UV-vis, as PCBM should have notable peak as C=O.
- 4) Follow the above mentioned question, the author should also include the working function comparison between SWCNTs w/ PCBM and w/o PCBM.
- 5) As PCBM is a widely utilized ETM and already reported in many monofacial PSCs using SWCNTs, why author should use additional SnO₂ ETM? I cannot understand the necessity of SnO₂. Maybe the author have already done this experiments, if so pls include this data.

Reviewer #2 (Remarks to the Author):

See attachment

Reviewer #2 Attachment on the following page

The paper shows highly efficient p-i-n perovskite solar cell using copper doped nickel oxide as HTL and PCBM as ETL, using single-walled carbon nanotubes (SWCNTs) as both front and back electrodes. The authors claim to have 98% bifaciality factor. The authors also made an analysis of the photovoltaic device showing its high cost efficiency. They claim to have the lowest weight per power due to the use of light weight SWCNT electrodes.

The main impact from the paper is the high bifaciality and bifacial power conversion efficiency. In bifacial devices the authors claimed to have 37% power conversion efficiency. This high efficiency is measured in a setup allowing 100% Albedo of AM1.5 as shown in Figure 3. This is equivalent to 1 Sun from the top and 1 sun from the bottom of the device.

Questions:

- How did you fabricate the SnO₂ layer on top of the perovskite layer?
- It is suggested to measure the PCE at different Albedos, since the condition of 100% Albedo is not practical?

Reviewer #3 (Remarks to the Author):

Zhang et al. reported bifacial single junction perovskite solar cells by using SWCNTs as transparent contacts. They assert an impressive bifacial power generation density of 36 mW/cm², accompanied by a high transparency in these films. However, it is crucial for the authors to elucidate the specific rationale behind their choice of SWCNTs for use in bifacial solar cells and make the focus of the manuscript on this specific topic. (please check the review paper as an example for the use of these materials in perovskite solar cells:

<https://doi.org/10.1016/j.matt.2021.12.011>)

The reported 36% cannot be construed as PCE (Pin/Pout), but rather represents a power generation density of 36 mW/cm² under specific albedo conditions. It is recommended to follow the guidelines outlined in Table 1 of the referenced paper (<https://doi.org/10.1016/j.joule.2022.05.014>) when presenting results for bifacial devices. In this context, albedo plays a critical role, and the generated power should be expressed with reference to specific albedo conditions. For instance, when denoting a power generation density (PGD) of 25 mW/cm² for BiFi200, it signifies that 25 mW of power is generated per square centimeter of device area under the simultaneous irradiance of 1-sun at the front (1000 W/m²) and 0.2-sun at the rear (200 W/m²).

1. Figure 1 can go to supporting information since this figure is well known. Also, the numbers in the figures are not easy to read please consider to change the colors and increase the fonts.
2. Page 5 line 109: please consider to change "bifacial factor" to "bifaciality factor"
3. What is the albedo value for 36.21? Please indicate in the text.
4. Page 9: Figure of merit (FOM) calculation for SWCNT films will be useful for the readers to compare the standard TCOs such as ITO and FTO. As a reference paper: <https://doi.org/10.1002/aelm.201600529>
5. In Figure 2b, the PL Intensity should be (normalized) not (a.u.)
6. Please correct the PCE to PGD in Figure 2d.
7. In Figure 2f-g-h, it is not clear what these normalized efficiencies represent. Please indicate in the figure (in addition to figure caption), what is the condition for stability tests. It will be easier to follow for the readers.
8. Page 15: The PL quenching for effective charge collection is a questionable discussion. At the interfaces, interfacial traps also reduce the lifetime and the PL intensity thus the Voc. In solar cells, this is not desirable. Therefore, it is not easy to conclude that PL quenching serves as a direct indicator of charge collection efficiency. Please check the following papers and adjust the discussions accordingly: <https://doi.org/10.1039/C7SE00603A> , DOI: 10.1126/science.abd4016
9. Page 20 line 405: Please correct as "350 to 600 nm"

Reviewer #4 (Remarks to the Author):

The authors showed the application of highly transparent and conductive single-walled carbon nanotubes (SWCNTs) to make bifacial perovskite solar cells, which displayed excellent bifacial properties and high power conversion efficiency. However, after careful consideration and assessment, I regret to notify you that I cannot accept your submission for publication. Please allow me to briefly summarize the primary rationale behind this decision:

1. In this study, the authors achieved the fabrication of bifacial perovskite solar cell devices using single-walled carbon nanotubes, resulting in a cost-effective material preparation approach. However, it's noteworthy that a similar approach was previously reported in the Energy Environ. Sci. journal in 2022. In light of this, what distinguishes the authors' investigation in terms of innovation?
2. The preparation method is inconsistent with the materials mentioned in the text and lacks precision. The discrepancy arises from the presentation of the device structure as glass/SWCNT@85%/Cu: NiOx /Perovskite/SnO₂/PCBM/SWCNT on Page 16, while the experimental methods fail to provide details on the preparation of SnO₂ on the perovskite layer. Moreover, the experiments did not utilize SnO₂ as the intended electron transport layer, highlighting an inconsistency between the proposed device structure and its actual implementation in the experiments.
3. The article posits that SWCNTs can achieve an exceptional 95% level of transparency. Please

provide substantiating evidence regarding the precise thickness of the carbon film under consideration. Moreover, when the resistance of SWCNTs closely approaches that of ITO, what constitutes the optimal thickness for this thin film? It would be of immense value if you could furnish corresponding photographic documentation, along with cross-sectional scanning electron SEM images, to enhance clarity and validate these assertions.

4. In Table S2, is the device performance represented by the highest efficiency or the average efficiency? If it's based on multiple experiments, please include the associated errors. Additionally, SWCNT@95% demonstrates improved transparency compared to SWCNT@90%, but it's noteworthy that J_{sc} current is lower while FF is higher. How can we interpret this phenomenon?

5. In Figure 3e, device testing was carried out with dual illumination from both the front and rear sides, resulting in a current of 40 mA cm^{-2} in the J-V curve. However, in practical scenarios, it is acknowledged that incident light intensity typically does not reach such levels. In the article, the device was illuminated with a single light source, resulting in a lower current of approximately 21 mA cm^{-2} . How can we increase the current output under realistic conditions? Additionally, what is the bandgap of the perovskite material?

6. Would carbon materials used as flexible electrodes not detach during the bending process?

Reviewer #1 (Remarks to the Author):

In this manuscript, Zhang et al., reported an interesting bifacial PSCs using highly transparent and conductive SWCNTs. Although the author delightedly reported a high PCE over 36%, this result was measured under the designated albedo factors and an active area of 0.09 cm², which somehow misleads the readers. It should be highlighted that the flexible device also showed excellent performance. Therefore, this work should be a milestone for the bifacial PSCs using SWCNTs, which definitely gives a general interest, and another way for future development of bifacial SWCNTs. This work could be accepted for publication in Nat. Commun. after revising the unclear points listed below:

Reply: Thank you for your positive and constructive feedback. We appreciate your suggestions and have revised the manuscript accordingly. Please see our detailed responses below.

Q1. It is true that SWCNTs have better optical transmittance. But it is inaccurate to mention the high cost using back metal electrodes compared with SWCNTs. The author should revise this description, e.g., in Abstract and Introduction part.

Reply: We thank the reviewer for the valuable suggestion. We have updated our description as follows:

Abstract: “However, the suboptimal optical transmittance of back metal electrodes together with the complex fabrication process associated with front transparent conducting oxides have hindered the development of efficient bifacial PSCs.”

Page 6: “Several solutions have been proposed to mitigate the damage caused by ITO or FTO deposition, including inserting a buffer layer on top of a perovskite layer or using damage-free sputtering approaches.²²⁻²⁴ These methods increase complexity, the number of required layers and the affordability of devices.”

“Other materials such as conductive polymers, metal grids, metal nanowires and carbon-based materials have also been proposed as alternatives to back electrodes in PSCs.²⁵⁻²⁸ Among these materials, the integration of carbon nanotubes (CNTs) into perovskite-based devices stands out owing to the exceptional electrical conductivity, stability, and mechanical flexibility of CNTs.^{26,29-33} These properties make them promising candidates for thermo-mechanical stable and flexible PSCs.”

Q2. In the discussion on working function of SWCNTs by UPS in Fig.S2, did author mean all SWCNTs films with different transmittance showed same working function? In addition, although SWCNTs electrodes mainly consisted of metallic tubes, there should remain some semiconductive tubes, which influences the transmittance and working function. Therefore, the author is recommended to conduct the UV-vis-NIR to check the metallic/semiconductive property by observing the M11, S11 etc., (doi.org/10.1016/j.carbon.2023.01.011).

Reply: We appreciate the reviewer's constructive feedback. To characterize the work function, we performed UPS measurements of SWCNT films with varying transmittances (Figure S4). The results revealed that the SWCNT film with the transmittance of 75%, 85%, 90% and 95%, has a similar work function of ~ 4.70 eV (measured 3 times for one SWCNT film). The work function is insensitive to the transmittance of the SWCNT film, in agreement with previous studies. (*Carbon* 2018, 130, 448e457).

Figure S4. UPS spectra of the SWCNT network with different optical transmittances and the calculated work function is ~ 4.70 eV.

Following the reviewer's suggestion, we conducted the UV-vis-NIR to study the metallic/semiconductive property of SWCNT film. Figure S3 displays the optical absorbance spectrum of the pristine SWCNT film. We detected the first and second semiconducting transitions of the van Hove singularities (E_{11}^S and E_{22}^S), as well as the first metallic transition of the van Hove singularity (E_{11}^M), indicating that the SWCNT film consisted of both metallic and semiconducting tubes. This is a common feature of SWCNT films prepared by a floating catalyst chemical vapor deposition (FCCVD) method, (*Adv. Funct. Mater.* 2021, 32, 2108541), and about 2/3 of the produced SWCNTs exhibit semiconducting behaviour. (*Adv. Funct. Mater.* 2019, 29, 1902273.)

Figure S3. The optical absorbance of the pristine SWCNT film.

We have amended our manuscript as follows:

On page 10: To investigate the electrical properties of these SWCNT films further, we performed optical absorbance spectroscopy and observed the first and second semiconducting transitions of the van Hove singularities (E_{11}^S and E_{22}^S), as well as the first metallic transition of the van Hove singularity (E_{11}^M) (Figure S3). Typically, the SWCNT films prepared by a floating catalyst chemical vapor deposition (FCCVD) method exhibit 1/3 metallic and 2/3 semiconducting characteristics.⁴⁴ The work function of the electrode is a key factor in enabling a high built-in voltage.⁴⁵ Therefore, the work function of SWCNT film was determined using ultraviolet photoelectron spectroscopy (UPS). The results indicate that the calculated work function of the SWCNT film is about -4.70 eV (Figure S4). We note that the work function of the SWCNT film, which is comparable to that of ITO, remains constant at approximately -4.70 eV) regardless of its optical transmittance.⁴⁶⁻⁴⁸

Q3. In Fig. 2b, the TEM showed that there is something (maybe surfactant) wrapped SWCNTs. Therefore, it is suspicious that Fig. S3 can conclude the PCBM surrounded the SWCNTs. If so, the author should collect more evidence to support Fig. S3, for instance compare by FTIR or UV-vis, as PCBM should have notable peak as C=O.

Reply: We thank the reviewer for the valuable suggestion.

To investigate the structure and composition of the SWCNT films, we performed TEM analysis. As shown in Figure S5a, the SWCNTs are interconnected by graphitic carbon, which was formed during the FCCVD process (see **Methods** section for details). We confirm that no surfactant was used during the FCCVD process. The graphitic carbon acts as a welding agent that enhances the electrical conductivity of the SWCNT films, as we have demonstrated in our prior works (*Sci. Adv.* 2018, 4, eaap9264; *J. Mater. Chem. A.* 2022, 10, 16986-16994). Figure S5b shows the TEM images of the PCBM-

coated SWCNT films, which reveal that the PCBM forms a thin layer on the surface of the SWCNT network.

Figure S5. TEM images of a) the SWCNT film and (scale bar = 50 nm) and b) the PCBM/SWCNT film (scale bar = 50 nm).

To further confirm the incorporation of PCBM onto the SWCNT film, the Fourier transform infrared spectroscopy (FTIR) was conducted. As shown in Figure S6, compared to the pristine SWCNT film, the PCBM/SWCNT film exhibits two prominent peaks, which are assigned to C=O and C-H, respectively. (*Chem. Mater.* 2020, 32, 5125–5133).

Figure S6. Fourier transform infrared spectroscopy (FTIR) transmission spectra of the SWCNT film and the PCBM/SWCNT film.

We have amended our statement on page 12:

The intimate contact between SWCNTs and PCBM was directly observed through TEM. The TEM analysis, as depicted in Figure S5, along with the Fourier transform infrared spectroscopy (FTIR) data shown in Figure S6, confirms the close interaction between SWCNTs and PCBM.^{39,50,51}

Q4. Follow the above mentioned question, the author should also include the working function comparison between SWCNTs w/ PCBM and w/o PCBM.

Reply: We thank the reviewer for the valuable suggestion. The UPS spectra of SWCNTs w/ PCBM and w/o PCBM are compared in Figure S9. The results indicate that the work function decreased from 4.70 eV to 4.55 eV when introducing PCBM. Therefore, PCBM can act as a n-type dopant for the SWCNT film, which was also confirmed by the Raman spectra in Figure S4 and the work reported by Maruyama et al. (*J. Phys. Chem. C*. 2017, 121, 25743–25749; *InfoMat*. 2019, 1, 559–570.)

Figure S9. UPS spectra of the SWCNT film and the PCBM/SWCNT film.

We then revised the statement on Page 13:

To further investigate the impact of PCBM on mixed SWCNTs, UV–vis spectra of SWCNT and PCBM-SWCNT were then analyzed. The results reveal a blue shift of the characteristic peak of SWCNTs, indicating strong π - π stacking interactions between PCBM and the SWCNT films (Figure S8).⁵⁷ And the work function of the SWCNT film decreased from ~ 4.70 eV to ~ 4.55 eV (Figure S9). Moreover, the resistance of the SWCNT film reduced from 28.8Ω to 21.9Ω upon contact with PCBM, presumably owing to the formation of π - π bonds between PCBM and SWCNTs, which facilitated electron transport (Figure S10).^{50,58}

Q5. As PCBM is a widely utilized ETM and already reported in many monofacial PSCs using SWCNTs, why author should use additional SnO₂ ETM? I cannot understand the necessity of SnO₂. Maybe the author have already done this experiments, if so pls include this data.

Reply: Thank you for your valuable comment. PCBM is a common material used in PSCs, but it also has some limitations, such as low thermal stability, poor ability to suppress ion migration, and severe self-aggregation, etc. (e.g. *J. Am. Chem. Soc.* 2022, 144, 12, 5400–5410; *Nano Energy*, 2019, 64, 103962; *Adv. Energy Mater.* 2022, 12, 2103567). These issues can be alleviated by combining PCBM with an inorganic SnO₂ layer, which can enhance the electron mobility of the hybrid SnO₂/PCBM layer and the operating stability of the device. We have previously reported this work in another publication (*Energy Environ. Mater.* 2023,0, e12595.)

Reviewer #2 (Remarks to the Author):

The paper shows highly efficient p-i-n perovskite solar cell using copper doped nickel oxide as HTL and PCBM as ETL, using single-walled carbon nanotubes (SWCNTs) as both front and back electrodes. The authors claim to have 98% bifocality factor. The authors also made an analysis of the photovoltaic device showing its high cost efficiency. They claim to have the lowest weight per power due to the use of light weight SWCNT electrodes. The main impact from the paper is the high bifocality and bifacial power conversion efficiency. In bifacial devices the authors claimed to have 37% power conversion efficiency. This high efficiency is measured in a setup allowing 100% Albedo of AM1.5 as shown in Figure 3. This is equivalent to 1 Sun from the top and 1 sun from the bottom of the device.

Reply: We are grateful to the reviewer for appreciating our work. Please find our detailed responses to each of the questions below.

Q1. How did you fabricate the SnO₂ layer on top of the perovskite layer?

Reply: We thank the reviewer for raising the question. We used spin-coated SnO₂ nanoparticles dispersed in butanol, which does not damage the underlying layers. The details of the SnO₂ fabrication have been added to the **Methods** section, and also can be found in our recent work published in *Energy Environ. Mater.* 2023,0, e12595.

Page 27: SnO₂ (2.5 wt%) in butanol solution was purchased from Avantama.

Page 31: The SnO₂ layer was prepared with the SnO₂ butanol solution, which was diluted with 1-butanol until the concentration reached 1.25 wt%. Subsequently, SnO₂ thin film was spin-coated on top of the perovskite with the diluted solution at 6000 rpm (2000 rpm/s) for 30 s and heated at 100 °C for 40 min.

Q2. It is suggested to measure the PCE at different Albedos, since the condition of 100% Albedo is not practical?

Reply: We are grateful for the reviewer's valuable suggestion. We acknowledge that the 100% albedo assumption is not realistic, and we have performed simulations under various albedo conditions that correspond to different environments (ranging from 96% for snowfields to 12% for tiles, Figure S13 and Figure S14). We have also measured the performance of all the devices under these conditions, and the results are presented in Figure 2d and Table S3.

Reviewer #3 (Remarks to the Author):

Zhang et al. reported bifacial single junction perovskite solar cells by using SWCNTs as transparent contacts. They assert an impressive bifacial power generation density of 36 mW/cm², accompanied by a high transparency in these films. However, it is crucial for the authors to elucidate the specific rationale behind their choice of SWCNTs for use in bifacial solar cells and make the focus of the manuscript on this specific topic. (please check the review paper as an example for the use of these materials in perovskite solar cells: <https://doi.org/10.1016/j.matt.2021.12.011>) The reported 36% cannot be construed as PCE (Pin/Pout), but rather represents a power generation density of 36 mW/cm² under specific albedo conditions. It is recommended to follow the guidelines outlined in Table 1 of the referenced paper (<https://doi.org/10.1016/j.joule.2022.05.014>) when presenting results for bifacial devices. In this context, albedo plays a critical role, and the generated power should be expressed with reference to specific albedo conditions. For instance, when denoting a power generation density (PGD) of 25 mW/cm² for BiFi200, it signifies that 25 mW of power is generated per square centimeter of device area under the simultaneous irradiance of 1-sun at the front (1000 W/m²) and 0.2-sun at the rear (200 W/m²).

Reply: We are grateful for the referee's insightful suggestion. We have carefully reviewed these works and incorporated the relevant changes into our manuscript.

Q1. Figure 1 can go to supporting information since this figure is well known. Also, the numbers in the figures are not easy to read please consider to change the colors and increase the fonts.

Reply: We thank the reviewer for their valuable advice. We have relocated Figure 1 to the supporting information (Figure S1), with enhanced readability and clarity.

Q2. Page 5 line 109: please consider to change “bifacial factor” to “bifaciality factor”

Reply: We are grateful for your helpful advice. We have changed the term ‘bifacial factor’ to ‘bifaciality factor’ in our manuscript.

Q3. What is the albedo value for 36.21? Please indicate in the text.

Reply: we thank the reviewer for this advice. We have added the following sentences to the manuscript:

On page 7: We adopt the BiFi notation to denote the corresponding PGD in a scientific and simple manner. For instance, a PGD of 25 mW cm^{-2} BiFi₂₀₀ implies that the device produces 25 mW per cm^2 under simultaneous irradiance of 1-sun ($1,000 \text{ W m}^{-2}$) at the front and 0.2-sun (200 W m^{-2} , denoted as BiFi₂₀₀) at the back.³⁷ We further evaluated the photovoltaic performance of these bifacial PSCs under various common environments with different albedo factors and the resultant PGD range from 16.75 mW cm^{-2} (tiles, BiFi₁₂₀) to 36.21 mW cm^{-2} (snowfield, BiFi₉₆₀).

Q4. Page 9: Figure of merit (FOM) calculation for SWCNT films will be useful for the readers to compare the standard TCOs such as ITO and FTO. As a reference paper: <https://doi.org/10.1002/aelm.201600529>

Reply: We thank the referee for the constructive advice. We have perused the paper and attempted to calculate the FOM values of our SWCNT films based on the suggested method.

We have added the following contents in our manuscript (on Page 9):

We also compared the figure of merit (FOM) values of these SWCNTs and found that the SWCNT@85% film presents the highest value (7.03×10^{-3}) among these SWCNTs (Figure S1), which is even higher than of ITO (5.07×10^{-3}).^{36,37}

Table S1. figure of merit values of SWCNT films with different optical transmittances.

	$R_{sh} (\Omega \text{ sq}^{-1})$	T(%)	FOM(%)
SWCNT@95%	138	95	0.434
SWCNT@90%	64	90	0.545
SWCNT@85%	28	85	0.703
SWCNT@75%	17	75	0.331

Q5. In Figure 2b, the PL Intensity should be (normalized) not (a.u.)

Reply: we thank the reviewer for raising this valuable point. We have updated the figure below.

Q6. Please correct the PCE to PGD in Figure 2d.

Reply: We thank the reviewer for pointing out this issue. We have modified the figure accordingly.

Q7. In Figure 2f-g-h, it is not clear what these normalized efficiencies represent. Please indicate in the figure (in addition to figure caption), what is the condition for stability tests. It will be easier to follow for the readers.

Reply: We are grateful for your helpful suggestion. Captions are added accordingly.

Q8. Page 15: The PL quenching for effective charge collection is a questionable discussion. At the interfaces, interfacial traps also reduce the lifetime and the PL intensity thus the Voc. In solar cells, this is not desirable. Therefore, it is not easy to conclude that PL quenching serves as a direct indicator of charge collection efficiency. Please check the following papers and adjust the discussions accordingly: <https://doi.org/10.1039/C7SE00603A>, DOI: 10.1126/science.abd4016

Reply: We appreciate the reviewer's question on the PL intensity shifts, which are a complex issue in perovskite solar cells, particularly when they involve the interactions between the perovskite layer and the electron-transporting material (ETM) or hole-transporting material (HTM). The reduction of the PL intensity can result from both nonradiative recombination and extraction of charge carriers from the active layers. In our device stacks, the latter is the dominant factor for the PL intensity reduction, as it is enhanced by the increased thickness of the SWCNT films. If the PL intensity reduction was due to the increased recombination rate, the PCE of the PSCs based on SWCNT@90%, SWCNT@85% and SWCNT@75% would be lower than that of the PSCs based on SWCNT@95%. However, our data (Table S3) show that the PCE increases with the SWCNT film thickness. Therefore, we conclude that the improved extraction of charge carriers from the active layers is the main reason for the PL intensity reduction. To make our statement clearer, we have updated the manuscript as follows:

On page 16: As a result, the decay time decreased from 11.39 ns to 4.61 ns as the optical transmittance of SWCNT films reduced from 95% to 75%, whereas the values were 9.63 ns and 6.72 ns for semi-devices with SWCNT@90% and SWCNT@85% respectively.⁶⁰ In conjunction with the PCE of resultant devices (Table S3), this indicated that thick SWCNT films could collect charges more efficiently than their thinner counterparts.^{37,61} In addition to the difference in conductivity of these SWCNT films, it is plausible that the compactness of the contact between PCBM and thick

SWCNT films may contribute to the observed reduction in carrier lifetime,⁶² which is consistent with the previous PL data.

Q9. Page 20 line 405: Please correct as “350 to 600 nm”

Reply: We thank the reviewer for this suggestion, and now the sentence has been corrected.

Reviewer #4 (Remarks to the Author):

The authors showed the application of highly transparent and conductive single-walled carbon nanotubes (SWCNTs) to make bifacial perovskite solar cells, which displayed excellent bifacial properties and high power conversion efficiency. However, after careful consideration and assessment, I regret to notify you that I cannot accept your submission for publication. Please allow me to briefly summarize the primary rationale behind this decision:

Reply: We thank the reviewer for the comment and agree that the performance of our bifacial devices is great. We have carefully read and analyzed the reviewer's comments and we have prepared a detailed response to each of them. We have also revised our manuscript accordingly to address the issues raised and to improve the clarity and quality of our presentation.

Q1. In this study, the authors achieved the fabrication of bifacial perovskite solar cell devices using single-walled carbon nanotubes, resulting in a cost-effective material preparation approach. However, it's noteworthy that a similar approach was previously reported in the Energy Environ. Sci. journal in 2022. In light of this, what distinguishes the authors' investigation in terms of innovation?

Reply: We appreciate the reviewer's question. The work published in Energy Environ. Sci. (referred to as the EES work) is of great importance for advancing the development of carbon-based PSCs. However, our work differs from the EES work in the following aspects:

- The general scope of our work. The EES work prepared PSC/CIGS tandem devices after making these carbon-based PSCs and characterized their performance. For our work, after making both rigid and flexible devices, we also performed a detailed analysis of the economic benefits of our carbon-based PSCs by simulating the annual power generation using our climate forecast model and calculating the manufacturing cost.
- The material and fabrication methodology. We propose a novel and simple fabrication methodology to develop low-cost and high-performance PSCs with all-carbon electrodes, which can eliminate the need for transparent conductive oxides (TCOs), such as fluorine-doped tin oxide (FTO) or indium tin oxide (ITO). Unlike the EES work that combined TCOs with carbon materials to make bifacial devices, which still require TCOs as the electrode.

We used SWCNT films as both the anode and cathode directly. Therefore, we removed the TCO layer entirely from the device structure, thereby reducing the cost and complexity of PSCs significantly. We demonstrated that our SWCNT-based PSCs have high power conversion efficiency and long-term stability, comparable to or even higher than those of the TCO-based PSCs. Therefore, we believe our device configuration can offer great potential for preparing efficient, long-term stable and low-cost PSCs.

Overall, we appreciate the recognition of our work by the other three reviewers, who have acknowledged its novelty and significance. We believe that our work contributes to the advancement of bifacial PSCs.

To highlight these innovation points, we have marked these contents in the manuscript as follows:

Page 7: 'Here, We demonstrate the fabrication of bifacial PSCs using highly transparent and conductive single-walled carbon nanotubes (SWCNTs) as both front and back electrodes without integrating with other materials. We investigate the interactions between SWCNTs and electron/hole transport materials (ETM/HTM) and their effects on the device performance.'

Page 8: 'Due to the excellent environmental durability of SWCNTs, these bifacial PSCs exhibit remarkable stability against moisture, light and heat conditions. We also simulated the power generation of these bifacial PSCs for the next 25 years. Moreover, SWCNTs are flexible and mechanically robust, which enables them to withstand repeated bending cycles. These features make SWCNTs suitable for flexible PSCs and enable new visions for the development of bifacial PSCs.'

Q2. The preparation method is inconsistent with the materials mentioned in the text and lacks precision. The discrepancy arises from the presentation of the device structure as glass/SWCNT@85%/Cu:NiOx /Perovskite/SnO₂/PCBM/SWCNT on Page 16, while the experimental methods fail to provide details on the preparation of SnO₂ on the perovskite layer. Moreover, the experiments did not utilize SnO₂ as the intended electron transport layer, highlighting an inconsistency between the proposed device structure and its actual implementation in the experiments.

Reply: We thank the reviewer for raising this important point. The SnO₂ layer is essential for extracting and transporting electrons in our structure, as it forms a hybrid SnO₂/PCBM layer that enhances the electron mobility of the ETM. This aspect has been discussed and studied in detail in our previous work (please see the reference *Energy Environ. Mater.* 2023,0, e12595). However, the layer that directly contacts the back SWCNT film is the PCBM, which is deposited on top of the SnO₂. Therefore, we focused on the interactions between the SWCNT and the PCBM layers, as they play a key role in transferring charges from ETM to SWCNT.

We have added the details of the SnO₂ layer in our manuscript as below:

Page 27: SnO₂ (2.5 wt%) in butanol solution was purchased from Avantama.

Page 31: The SnO₂ layer was prepared with the SnO₂ butanol solution, which was diluted with 1-butanol until the concentration reached 1.25 wt%. Subsequently, SnO₂

thin film was spin-coated on top of the perovskite with the diluted solution at 6000 rpm (2000 rpm/s) for 30 s and heated at 100 °C for 40 min.

Q3. The article posits that SWCNTs can achieve an exceptional 95% level of transparency. Please provide substantiating evidence regarding the precise thickness of the carbon film under consideration. Moreover, when the resistance of SWCNTs closely approaches that of ITO, what constitutes the optimal thickness for this thin film? It would be of immense value if you could furnish corresponding photographic documentation, along with cross-sectional scanning electron SEM images, to enhance clarity and validate these assertions.

Reply: We appreciate the reviewer's questions. We concur with the reviewer that the precise thickness of these SWCNT films is a key factor for the overall performance of PSCs. For our SWCNT network, however, the SWCNT films consist of randomly cross-stacked SWCNTs, which leads to a rougher morphology than that of ITO/FTO. As revealed by our AFM data (Figure R1a), the average thickness of SWCNT@95% film is about 11 nm. In addition, the cross-sectional SEM (Figure R1b) reveals the thickness variation of SWCNT@95% from ~8 to ~14 nm, in general agreement with AFM observation. Therefore, we can only provide an average value ($\sim 11 \pm 3$ nm) rather than a precise value for the SWCNT@95% films.

For the SWCNT film with a resistance comparable to that of ITO, namely SWCNT@85%, we performed the cross-sectional SEM of a Glass/SWCNT@85%/Perovskite stack. The SWCNT layer had an approximate thickness of 23 nm for SWCNT@85% (Figure R1c, d). All these data are in agreement with our previous report (*J. Mater. Chem. A*, 2022,10, 16986-16994).

Figure R1. a) AFM image of the SWCNT@95% film. b) cross-sectional SEM of the glass/SWCNT@95%/Perovskite stack (scale bar=100nm). c) cross-sectional SEM of the glass/SWCNT@85%/Perovskite stack (scale bar=100nm).d) cross-sectional SEM of the glass/SWCNT@85%/Perovskite/SWCNT@85% stack (scale bar=100nm).

We have added the corresponding thickness values to our revised manuscript:

On page 9: To balance the sheet resistance and optical transmittance, we increased the thickness of SWCNTs and prepared SWCNT@90%, SWCNT@85% and SWCNT@75% films (~8 nm for SWCNT@95%, ~15 nm for SWCNT@90%, ~21 nm for SWCNT@85% and ~28 nm for SWCNT@75%),³⁹ and we found that the SWCNT@75% film has a sheet resistance comparable to that of the ITO film.

Q4. In Table S2, is the device performance represented by the highest efficiency or the average efficiency? If it's based on multiple experiments, please include the associated errors. Additionally, SWCNT@95% demonstrates improved transparency compared to SWCNT@90%, but it's noteworthy that Jsc current is lower while FF is higher. How can we interpret this phenomenon?

Reply: We acknowledge the referee for bringing up these questions. The performance distribution errors have a great reference value. Hence, we have updated Table S3 as below according to the referee's suggestions.

Table S3. Photovoltaic parameters of bifacial PSCs with a configuration of Glass/SWCNT@85%/Cu:NiO_x/Perovskite/SnO₂/PCBM/SWCNT (various optical transmittance), illuminated from the front and back side (15 devices were measured under each condition).

	Direction	V _{oc} (V)	J _{sc} (mA cm ⁻²)	FF (%)	PCE (%)	BiFi (%)
SWCNT@95%	Front	1.08±0.03	18.22±1.55	76.28±2.31	15.07±1.21	93.36
	Back	1.07±0.04	17.21±1.62	76.37±2.50	14.01±1.15	
SWCNT@90%	Front	1.13±0.02	20.57±1.34	74.99±2.28	17.41±1.16	94.25
	Back	1.12±0.03	20.61±1.28	71.26±2.11	16.41±1.05	
SWCNT@85%	Front	1.13±0.02	20.95±1.25	78.49±2.33	18.54±1.12	98.27
	Back	1.13±0.02	20.18±1.16	79.21±2.22	18.22±1.04	
SWCNT@75%	Front	1.13±0.03	21.89±1.18	77.80±2.41	19.14±1.11	74.24
	Back	1.13±0.04	17.01±1.45	73.79±3.17	14.21±1.63	

Besides, the Table S5 also has been amended.

Table S5. Photovoltaic parameters of a flexible bifacial PSC with a configuration of SWCNT@85%/Cu:NiO_x/Perovskite/SnO₂/PCBM/SWCNT@85%. (15 devices were measured under each condition)

Illumination direction	J-V direction	V _{oc} (V)	J _{sc} (mA cm ⁻²)	FF (%)	PCE (%)
Front side	Forward	1.13±0.02	19.63±0.55	77.11±1.32	17.13±0.43
	Reverse	1.12±0.01	19.44±0.48	76.79±1.25	16.80±0.37
Back side	Forward	1.12±0.02	19.39±0.61	75.77±1.14	16.60±0.34
	Reverse	1.11±0.02	19.17±0.54	75.79±1.06	16.21±0.30

For the J_{sc} , the main reason for the lower J_{sc} of SWCNT@95%-based PSCs compared with the SWCNT@90%-based counterparts is the high resistance of the SWCNT@95% films (Figure 1d, 138 Ω sq⁻¹), which is much higher than that of the SWCNT@90% films (Figure 1d, 64 Ω sq⁻¹). Consequently, the J_{sc} loss is severe in SWCNT@95%-based PSCs. For the FF, as shown in Figure R1a,b. The SWCNT@95% films are not uniformly distributed, resulting in some areas being composed of dense and compact SWCNTs while some areas only contain a few SWCNTs, which agrees with the reply to Q3. In this case, some thin SWCNT areas, e.g. ~8 nm thick areas, are only composed of about 4 tubes (the diameter of each tube is ~2 nm, Figure S2), resulting in a relatively limited conductivity. This is also reflected by the large variation in the performances of the corresponding devices. To demonstrate this, we have summarized the photovoltaic parameters of 25 SWCNT@95%-based PSCs and 15 SWCNT@90%-based PSCs in Figure R2. The reason why we use 25 SWCNT@95%-based PSCs is that their performances fluctuate significantly, as shown in Figure R2 a-d. We selected the top 15 representative devices with a narrow distribution out of 25 devices to calculate the average value (in the red circle), which shows that the SWCNT@95%-based PSCs have a slightly higher FF than the SWCNT@90%-based PSCs. On the contrary, the 15 SWCNT@90%-based PSCs do not show much fluctuation (Figure R2 a-d).

Figure R2. Device performance statistics (15 devices for each device type) for perovskite solar cells with different SWCNT films. (a) V_{oc} , (b) J_{sc} , (c) FF, and (d) PCE. Device parameters are obtained from reverse J-V scans.

Q5. In Figure 3e, device testing was carried out with dual illumination from both the front and rear sides, resulting in a current of 40 mA cm⁻² in the J-V curve. However, in practical scenarios, it is acknowledged that incident light intensity typically does not reach such levels. In the article, the device was illuminated with a single light source, resulting in a lower current of approximately 21 mA cm⁻². How can we increase the current output under realistic conditions? Additionally, what is the bandgap of the perovskite material?

Reply: We agree with the referee's comment on the practical application of our device. Indeed, the dual illumination from both the front and rear sides is an ideal condition that may not be easily achieved in reality. We believe that our device, however, still has great potential for improving the current output under realistic conditions. There are several possible ways to achieve this goal: 1) Devices can be deployed to areas where albedo is high (e.g. snowfield (the albedo is around 96%, which is very close to the 100%, as demonstrated by NREL: <https://www.osti.gov/servlets/purl/1645703>), urban areas where glasses are heavily deployed, the sunlight reflected by glasses can help bifacial PSCs capture more sunlight. 2) Installing artificial mirrors/reflectors to these bifacial PSCs to enhance the sunlight reflection to the bifacial PSCs. (*Journal of Cleaner Production*, 397, 2023, 136541).

The bandgap of our perovskite is 1.62 eV as calculated by UPS (Figure R3), which agrees with our previous work (*Energy Environ. Mater.* 2023,0, e12595).

Figure R3. UPS data of the perovskite active layer, and the bandgap of the perovskite layer is 1.62 eV.

Q6. Would carbon materials used as flexible electrodes not detach during the bending process?

Reply: We express our gratitude to the referee for asking the question. The electrode detachment is a critical issue for flexible devices, especially for devices that will be bent, twisted or folded repeatedly. Thus, we have done the bending test of our all-carbon-electrode-based flexible PSCs, which survived and maintained ~97% of their original performances without detachment after 1000 times bending (Figure 3d), no detachment was observed during bending, which is far better than the ITO-based counterparts. This result is also in agreement with our previous work (*Adv. Funct. Mater.* 2021, 31, 2104396).

Reviewer #1 (Remarks to the Author):

I think that the authors have adequately addressed the comments made by the reviewers in the revised version of the manuscript. Therefore, I have no further comments.

Reviewer #2 (Remarks to the Author):

The author's replies are satisfactory regarding the first question, concerning the preparation of SnO₂ layer. The author added the preparation procedure in the Methods section.

Regarding the second question concerning the measurement at different albedo conditions:

It was vital to verify the measurement set up indicated in figure 3 which corresponds to 100% Albedo. Which is 1 sun AM 1.5 at both top and rear sides of the solar cell.

The authors justified their results by making simulation at different albedo conditions corresponding to different fields. "(ranging from 96% for snowfields to 12% for tiles, Figure S13 and Figure S14)".

It is still arguable that these measurement conditions could occur in reality, even in snow fields, where the albedo could reach 93%.

In general, I suppose the authors replies are considered satisfactory. and considering the new modifications in the manuscript I think it is good to be published in Nature Communication.

Reviewer #3 (Remarks to the Author):

The authors have addressed all the comments, the manuscript is acceptable for acceptance. I only have a small correction about PCE. For bifacial devices, power generation density is a better term to be used. So, I proposed to use power generation density (PGD) instead of bifacial power conversion efficiency in the manuscript.

Reviewer #4 (Remarks to the Author):

As a reviewer, I appreciate the authors' thorough response. They have not only addressed our previous concerns in detail but have also supplemented their study with relevant characterization data, which enhances the credibility and depth of their research. However, I still have some queries that need to be addressed to meet the publication standards.

1: We have observed that the thickness difference between the SWCNT@85% and SWCNT@95% carbon layers is not significant, and both are quite thin. In terms of experimental operations, how can we reasonably control this? Additionally, due to the thinness of the carbon layer and its uneven distribution, there is significant performance fluctuation, as shown in Figure S2. How can we ensure the reproducibility of the experiment?

2: while the response includes quantitative data on the bending test performance, it could benefit from additional details such as:

The conditions under which the bending tests were performed (e.g., bending radius, speed, and environmental conditions).

Information on how the bending affects the microstructure of the carbon material could be essential for understanding the long-term durability of the electrodes.

In addition, what is the thickness of the carbon layer of all-carbon-electrode-based PSCs in Figure 2g? How is the transparency?

Reviewer #1 (Remarks to the Author):

I think that the authors have adequately addressed the comments made by the reviewers in the revised version of the manuscript. Therefore, I have no further comments.

Reply: Thank you for your positive feedback on our revised manuscript. We appreciate your recognition of the quality and standards of our work.

Reviewer #2 (Remarks to the Author):

The author's replies are satisfactory regarding the first question, concerning the preparation of SnO₂ layer. The author added the preparation procedure in the Methods section.

Regarding the second question concerning the measurement at different albedo conditions:

It was vital to verify the measurement set up indicated in figure 3 which corresponds to 100% Albedo. Which is 1 sun AM 1.5 at both top and rear sides of the solar cell. The authors justified their results by making simulation at different albedo conditions corresponding to different fields. "(ranging from 96% for snowfields to 12% for tiles, Figure S13 and Figure S14)".

It is still arguable that these measurement conditions could occur in reality, even in snow fields, where the albedo could reach 93%.

In general, I suppose the authors replies are considered satisfactory. and considering the new modifications in the manuscript I think it is good to be published in Nature Communication.

Reply: We thank the reviewer for the comments which have improved the quality and scope of this study. In this work, we intend to investigate the albedo effect on the performance of perovskite solar cells (PSCs). To do so, we need a reliable albedo dataset that covers a wide area and has a similar latitude to London, UK. We found such a dataset from Fort Peck, Montana, recorded by NASA and DuraMAT, as shown in Figure R1. The albedo value could reach as high as 95.5%, hence, we believe ~96% is a reasonable value.

Moreover, we note that a small albedo difference (e.g., concrete vs. tile, around 6% difference) does not have a significant impact on the performance of our bifacial PSCs. We emphasize the novelty of our bifacial PSCs, which use all-carbon electrodes, is that they can efficiently harvest the albedo energy and generate more power per unit area than conventional solar cells.

Figure R1. Albedo data of Fort Peck, Montana, USA from 2003 to 2019.

Reviewer #3 (Remarks to the Author):

The authors have addressed all the comments, the manuscript is acceptable for acceptance. I only have a small correction about PCE. For bifacial devices, power generation density is a better term to be used. So, I proposed to use power generation density (PGD) instead of bifacial power conversion efficiency in the manuscript.

Reply: We appreciate your positive feedback on our revised manuscript. Following your suggestion, We have substituted the term 'PCE' with 'PGD' in our revised manuscript and supporting information when discussing the bifacial devices.

Reviewer #4 (Remarks to the Author):

As a reviewer, I appreciate the authors' thorough response. They have not only addressed our previous concerns in detail but have also supplemented their study with relevant characterization data, which enhances the credibility and depth of their research. However, I still have some queries that need to be addressed to meet the publication standards.

Reply: We thank the reviewer for their valuable insight and input on this work, and greatly appreciate their recognition of our research.

Q1: We have observed that the thickness difference between the SWCNT@85% and SWCNT@95% carbon layers is not significant, and both are quite thin. In terms of experimental operations, how can we reasonably control this? Additionally, due to the thinness of the carbon layer and its uneven distribution, there is significant performance fluctuation, as shown in Figure S2. How can we ensure the reproducibility of the experiment?

Reply: We thank the reviewer for pointing out these observations. To produce these films, we used the floating catalyst chemical vapor deposition (FCCVD) method, and adjusted the collection time to control the thickness. We then tested the reproducibility of these films by preparing and measuring 10 samples for each level of optical transparency (95%, 90%, 85%, 75%). The results are displayed in Table R1. The narrow distribution of thickness indicates good reproducibility of SWCNT films.

Table R1. Thickness statistics of SWCNTs with different optical transmittance (10 samples for each optical transmittance).

	SWCNT@95%	SWCNT@90%	SWCNT@85%	SWCNT@75%
Thickness	7.6±1.7nm	15.2±1.3nm	21.4±1.4nm	28±1.2nm

As for the performance on the device level, we measured the photovoltaic parameters of four sets of devices, each with a different SWCNT concentration (95%, 90%, 85%, and 75%) as the top electrode, and 15 devices per set. Figure R2 shows that, except for the set of devices with 95% SWCNT which contains outliers, the other three sets exhibit good reproducibility. In our work, the devices with SWCNT@85% demonstrate superior bifacial performance among all the sets, making them the ideal choice for bifacial PSCs application.

Figure R2. Device performance statistics (15 devices for each set) for PSCs with different SWCNT films (measured from the front side).

Q2: while the response includes quantitative data on the bending test performance, it could benefit from additional details such as:

The conditions under which the bending tests were performed (e.g., bending radius, speed, and environmental conditions).

Information on how the bending affects the microstructure of the carbon material could be essential for understanding the long-term durability of the electrodes.

In addition, what is the thickness of the carbon layer of all-carbon-electrode-based PSCs in Figure 2g? How is the transparency?

Reply: We thank the reviewer for the constructive suggestions. We have amended the description of the bending test in the manuscript.

On page 22: We performed a mechanical stability test on the flexible PSCs with all-carbon electrodes and the control devices by bending them with a 4 mm radius at a rate of 20 cycles/min in a glovebox. We measured the PCE every 100 cycles for a total of 1000 cycles.

To investigate the effect of bending on the microstructure, we measured the sheet resistance (R_s) and performed scanning electron microscopy (SEM) on both PEN-ITO and PEN-SWCNT before and after the bending test. The R_s value of SWCNT electrodes remains stable after 1000 cycles, demonstrating their high mechanical stability (Figure S21). However, the R_s value of commercial flexible ITO substrates increases by almost 2000 times (Figure S21). The SEM images of ITO (Figure S20) show clear cracks after bending with a 4 mm radius at a rate of 20 cycles/min, which accounts for the degradation of flexible ITO substrates. In contrast, the SWCNT films (using SWCNT@85%) retain their morphology after 1000 bending cycles (Figure S22 a,b).

Figure S21. Variations in sheet resistance increase of the PEN-ITO and PEN-SWCNT@85% as a function of the cycles of bending.

Figure S22. SEM images of SWCNT@85% a) before and b) after 1000 times bending.

Action:

We have revised our description of the bending test on page 22:

'The control device showed severe degradation, with the ITO cracking or breaking after 1000 bending cycles (Figure S20), leading to a drastic rise in sheet resistance (Figure S21). Conversely, the flexible double-sided SWCNT@85% PSC maintained over ~97% of its initial PGD (Figure 3d), no obvious change on the morphology of SWCNT was observed (Figure S22 a,b), and the sheet resistance remains constant (Figure S21). Therefore, SWCNT films unlock a promising approach to achieve lightweight, highly-efficient, mechanically robust and long-term stable flexible bifacial PSCs.'

And we also added Figure S21 and Figure S22 to the supporting information.

Figure 2g illustrates the device structure of the double-sided SWCNT@85% PSC (SWCNT@85%/HTM/Perovskite/ETM/SWCNT@85%). The transparency of the SWCNT@85% electrodes is around 85% (Figure 1c), the thickness of the SWCNT@85% is around 22nm, and the optical transmittance of the whole device is around 17.6% (Figure 3b).